# Text speaks louder: Insights into personality from natural language processing

**David Saeteros**[1], **David Gallardo-Pujol**[1,2,3], **Daniel Ortiz-Martínez**[4]*

**1** Department of Clinical Psychology and Psychobiology, Universitat de Barcelona, Barcelona, Spain,
**2** Institut de Neurociències (UBNEURO), Barcelona, Spain, **3** Institute of Complex Systems (UBICS),
Barcelona, Spain, **4** Department of Mathematics and Computer Science, Universitat de Barcelona,
Barcelona, Spain

* daniel.ortiz@ub.edu

INDIA

**Peer Review History:** PLOS recognizes the
benefits of transparency in the peer review
process; therefore, we enable the publication of
all of the content of peer review and author
responses alongside final, published articles.
The editorial history of this article is available
here: https://doi.org/10.1371/journal.pone.
0323096

**Data availability statement:**
https://osf.io/sgvta/

## Abstract

In recent years, advancements in natural language processing (NLP) have enabled new
approaches to personality assessment. This article presents an interdisciplinary inves-
tigation that leverages explainable AI techniques, particularly Integrated Gradients, to
scrutinize NLP models' decision-making processes in personality assessment and ver-
ify their alignment with established personality theories. We compare the effectiveness
of typological (MBTI) and dimensional (Big Five) models, utilizing the Essays and MBTI
datasets. Our methodology applies log-odds ratio with Informative Dirichlet Prior (IDP)
and fine-tuned transformer-based models (BERT and RoBERTa) to classify personality
traits from textual data. Our results demonstrate moderate to high accuracy in person-
ality prediction, with NLP models effectively identifying personality signals in text in line
with previous studies. Our findings reveal theory-coherent patterns in language use asso-
ciated with different personality traits, while highlighting important biases in the MBTI
dataset that yielded less robust results. The study underscores the potential of NLP in
enhancing personality psychology and emphasizes the need for further interdisciplinary
research to fully realize the capabilities of these transparent technologies.

## Introduction

Language is one of the most fascinating tools human beings have invented. Researchers in
psychology have regarded it as a source of insights into the inner world, leading to ground-
breaking discoveries about the interplay between language use and psychological character-
istics. From Freud's controversial analysis of slips of the tongue [1], through the use of self-
reference pronouns to detect psychological phenomena [2–4] until more recent computa-
tional approaches, the field has evolved significantly [5].

The interest derived from language as a window to psychological features led into the
*words-as-attention* premise that posits verbal behavior as a reflection of what someone is pay-
ing attention [6,7], and push the research into studying personality [8], life experiences [9],
and even cultures [10], and societies [11].

**Funding:** This research was supported by the Government of Catalonia (https://govern.cat/gov/), Grant number 2021SGR0709 received by DGP, and by the Ministry of Science and Innovation of Spain, MCIN/AEI/10.13039/501100011033 (https://www.ciencia.gob.es/), Grant number PID2020-119755GB-I00 received by DGP. The funder had no role in the study design, data collection and analysis, decision to publish, or preparation of the manuscript.

**Competing interests:** The authors have declared that no competing interests exist.

The exploration of this linguistic-psychological landscape was significantly advanced by the development of the Linguistic Inquiry Word Count (LIWC) [12]. This algorithm compared input words with a predetermined dictionary, calculating proportions of word categories to infer psychological characteristics. This would finally lead to computing correlations with psychological and social outcomes [3].

Despite its limitations, particularly in contextual understanding, LIWC demonstrated significant correlations with various outcomes in the field of psychology [3,13–17], paving the way for more advanced linguistic analysis in psychology [18]. This approach aligned with the lexical hypothesis—the foundational idea that important personality characteristics become encoded in language [19]—and suggested that computational analysis of natural language could reveal meaningful psychological patterns.

With the introduction of new AI tools, we have seen studies with a dedicated interest to topics from the social sciences. In particular, many studies aim to capture the signal from personality traits (e.g. the Big Five), employing LLMs, in labeled databases of texts (see Tables 1 & 2 in Supplemental Materials 1, henceforth referred to as S1 File). The lexical hypothesis—which posits that important personality characteristics become encoded in language [19]—provides theoretical grounding for these studies. These studies serve to bridge the gap between the original lexical hypothesis and the world of natural language production. Most of these results, presented later in the Introduction, have performed highly at classifying texts according to personality labels.

These promising results have sparked discussion about the relative merits of computational versus traditional approaches to personality assessment [31]. Traditional Personality Assessment (TPA) and NLP-based personality assessment (NLP/PA) represent distinct approaches with their own advantages and limitations [89]. While TPA relies on questionnaires and self-reported data from typically smaller samples, NLP/PA can analyze vast amounts of naturalistic language data, potentially offering greater representativeness and reduced subjective bias [31,78]. However, the field is still establishing how accurately personality can be inferred from spontaneous linguistic expression [62]. NLP/PA's premise that personality manifests in natural language behavior [5] requires further validation, even as it opens new possibilities for understanding how people think, feel, and organize their world [3].

It is also important to highlight that even if NLP/PA can analyze sources of spontaneous texts, this has no *a priori* overlap with the lexical hypothesis. Some researchers have wrongly conceptualized the lexical hypothesis as the personality-related information that can be found in everyday languages [22,70,90], when in fact it is something deeper than that, namely that every relevant psychological individual difference —including personality— should be found in some way codified in human languages, in a representative and exhaustive corpus of any given language [19,20]. Recent applications of this approach for the study of personality traits in African and Japanese populations can be found in Thalmayer et al. [87] and Hashimoto et al. [41] respectively.

Building on this understanding of the lexical hypothesis, we claim that the NLP/PA still needs to reveal its full potential in studying psychological variables through text analysis. Most current applications can be divided into two groups: the first includes studies that use textual data to predict personality attributes, either traits or types [18,34,49,52,55,68,69,73,83, 92,96].

The second group is exploring novel applications like modeling and predicting a wide range of traits, scales, and constructs [22,31], using chat-bots for personality assessment [35], inferring personality questionnaire item-responses from texts [93], guiding LLMs to respond according to a specific personality configuration [76], generating items for personality questionnaires [43,44], exploring the overlap between the semantic representation of LLM and

personality data generated from questionnaires [31], and the field holds great promises for the following years. These innovative approaches demonstrate the potential of NLP/PA to not only complement traditional methods but also to open up new avenues for personality research and assessment.

In the first group of articles many text datasets are available online [56,86,96]. Three datasets, in particular, have been extensively explored using LLM and neural networks: the Facebook data from myPersonality project [48,81] —no longer available for research purposes—, the Essays dataset [64], and the MBTI dataset [47].

The MBTI dataset is of particular interest due to its basis on the Myers-Briggs Type Indicator (MBTI), a measure that, while highly popular online, has questionable validity in the field of personality psychology [54]. This dataset comprises authors' posts from a personality psychology online forum that brings together individuals interested in the topic. The most evident bias in this dataset is the authors' prior knowledge of the MBTI theory and their own types. A more detailed explanation of this dataset is provided in the Methods section.

Tables 1 and 2 in S1 File summarize the studies that have explored both the Essays and the MBTI dataset, focusing on the accuracies or similar metrics achieved in the classification tasks. As is evident from these results, improving methodological accuracy has been a primary objective of these studies. To this aim, researchers have employed a varied range of methods, from traditional machine learning (e.g., SVM, Random Forest) to more advanced neural network architectures (e.g., BERT, CNN). The accuracies found vary considerably across studies and traits/types, ranging from just above chance (around 0.5) to high accuracy (above 0.8). A notable observation is the consistent analysis of the MBTI dataset, which can be explained by its huge popularity online and the predominance of computer scientists over psychologists among researchers. This trend persist despite extensive criticism of MBTI from personality psychologists. Moreover, accuracies for MBTI types (Table 2 in S1 File) often appear higher than those for Big Five traits (Table 1 in S1 File), which may reflect the dataset's bias rather than superior predictive power of the MBTI model.

This persistent focus on the MBTI dataset, despite its known limitations and the higher accuracies it yields, highlights a potential disconnect between computer science and personality psychology. The emphasis on improving accuracy metrics, as noted in the studies summarized, may sometimes come at the expense of theoretical validity. This underscores the need for interdisciplinary collaboration in the new applications as a necessary step for a more steadfast advancement of psychology and for avoiding what Boyd and Schwartz [25] described as the "square peg into a round hole" problem.

For the second group of articles, researchers have to get more creative. One of the most relevant efforts in this group is the work published by Cutler and Condon [31]. In three separate studies, they introduce useful practices and concepts to bridge the gap between NLP and personality assessment; such as the idea that personality traits are word vectors [30] or the idea of testing the conceptual and functional overlap of personality terms studied with psychometrics and NLP techniques. These studies show great promise and should be pursued further in future research.

The field's singular focus on improving accuracy or other performance metric sidesteps the question of what drives the model's decision. Moving forward, it's crucial to balance the pursuit of improved accuracy with critical evaluation of the underlying constructs and data quality. Specifically, we need to determine whether the accuracy values are really obtained from personality signals in the data or if they result from noise, overfitting or model hallucinations. Explainability techniques can help researchers understand how these models make decisions [74], ensuring that high accuracy is not achieved at the expense of theoretical validity or interpretability.

This approach is crucial for an effective advancement of the field, given that some researchers have started to advocate for a replacement of TPA with NLP/PA [34], and this claim might be supported by the data only by biases inherent to the model performance. To address this, personality psychology should incorporate explainability algorithms, already widely used in NLP tasks [74]. This will allow us to analyze how individuals with different personality traits or types use language, providing new insights into the relationship between personality and linguistic expression.

Our study leverages state-of-the-art NLP models to examine how personality traits and types manifest in natural language. While we implement classification models for the Big Five and MBTI labels from the Essays and MBTI datasets respectively, our primary focus is not on prediction performance. Instead, we use explainable AI techniques to analyze how these models process personality-relevant information in text, examining whether their decision-making patterns align with established personality theories. This approach allows us to study personality expression in language at an unprecedented level of detail, offering new insights into how different traits and types are reflected in natural writing. Through this combination of advanced NLP techniques and explainable AI, we aim to deepen our understanding of the relationship between personality and language use.

1. **The accuracy for classification of the Big Five traits on the Essays dataset will range from low to moderate due to the nature of the input data to the classifier.** H1 stems from the characteristics of the Essays dataset. While it provides a balanced distribution across categories, it contains stream-of-consciousness writing that may not always clearly reflect personality traits. This type of spontaneous, unstructured text can be challenging for classification algorithms, accounting for a moderate rather than perfect accuracy.

2. **The accuracy for MBTI types will not reflect its true predictive power due to the unbalanced nature of the MBTI dataset. The Area Under the Curve (AUC) will provide a more reliable measure of model performance.** Because of this, the accuracy will be higher for the MBTI dataset than for the Big Five, and the comparison of accuracy scores between the Big Five and MBTI datasets is inappropriate.

3. **The words most influential in Big Five classification will be theory coherent with the Big Five traits content.** This hypothesis is grounded in the theoretical foundation of the Big Five model. As it is based on empirically derived personality traits that are thought to be reflected in natural language use, we expect the words most influential in classification to align with the theoretical content of these traits. Given that the classification performance is not perfect, we also expect to find a few words with no coherence with the theory.

4. **In the MBTI model, the most influential words will include a high frequency of the MBTI categories, reflecting the dataset's bias towards explicit discussion of MBTI concepts rather than natural language patterns associated with personality.** The hypothesis arises from the nature of the MBTI dataset, which consists of posts from users who are aware of their MBTI classification and are discussing personality topics. This self-awareness and topical focus may lead to frequent use of MBTI-related terminology, potentially stemming from the MBTI's limitations as a robust personality model.

Although we advocate for more second-group studies, this study is relevant and increasingly perceived as a necessity by psychologists [32]. Our research makes several novel contributions to the field. First, we assess the robustness of the Big Five model across diverse data treatments [75], in this case NLP techniques. Second, we provide the first comprehensive explainability analysis of personality detection models, demonstrating clear theory coherence between Big Five traits and model attention patterns. Third, we uncover and quantify the impact of self-reference effects in MBTI classification, challenging previous performance metrics. By exploring our LLM's decision-making processes using explainability algorithms [74],

we address ongoing discussions about model evaluation [27]. This approach aligns with substantive validity recommendations outlined by Bleidorn and Hopwood [24] and contributes to improving personality assessment as a key element in social change [38].

## Materials and methods

This study employs a multi-faceted approach to explore personality through natural language analysis. Our methodology combines traditional NLP techniques with advanced NLP models and explainable AI methods. We begin with data preprocessing of two distinct datasets. We then apply various analytic techniques, including log-odds ratio analysis, fine-tuned language models, and integrated gradients for model interpretation. Our evaluation uses both standard classification metrics and novel approaches to visualize and interpret model decisions. This comprehensive approach allows us to not only predict personality traits from text but, more importantly, to understand the linguistic features driving these predictions.

### Data collection and preprocessing

This study employs two datasets to explore personality through natural language analysis. The first dataset, known as the stream-of-consciousness or the Essays dataset, was compiled by Pennebaker and King [64] and annotated by Mairesse et al. [51]. It employs the binary classification methodology outlined by Oberlander and Nowson [60]. The second dataset originates from the MBTI forum and includes labels based on the Myers-Briggs Type Indicator (MBTI).

The Essays dataset consists of a collection of essays gathered between 1997 and 2004 where participants were prompted to write for 20 minutes about whatever came to mind. It contains 2479 essays totaling approximately 1.9 million words (Table 3 in S1 File). Each essay is associated with binary labels (1 or 0) representing the presence or absence of each of the Big Five traits, namely Openness to Experience, Conscientiousness, Extraversion, Agreeableness, and Neuroticism. These labels were derived from participants' self-reports on the Five Factor Inventory [46].

The MBTI dataset, publicly available on Kaggle, comprises information from 8600 individuals who posted on the Personality Café forum. Each individual is associated with 50 forum posts, separated by three pipe characters ("|||"), and categorized into one of the 16 MBTI types. We augmented this dataset by adding columns representing each pair of MBTI types (I/E, N/S, T/F, J/P) with binary values assigned based on the presence of specific MBTI indicators (e.g. E, S, T, P) (Table 4 in S1 File), following similar practices in the literature. This dataset serves as an alternative perspective on personality assessment albeit with acknowledged limitations regarding its psychometric validity.

Due to the fact that both the Essays and the MBTI datasets are composed of a collection of long texts that exceed the word limit of the models used to process them (typically, the maximum word length is 512 words), we created modified versions where those long texts were divided into fragments of 256 words. This treatment allowed us to take advantage of all the information contained in the datasets, in contrast to the approach adopted in other works, where the texts were simply truncated to the word limit imposed by the models.

As it was explained in the introduction, the MBTI dataset participants were aware of their classification, which may introduce an artificial bias in the content of the texts. Indeed, a visual analysis of some of them clearly showed that, on many occasions, the participants were openly discussing their own classification or that of other participants, writing the individual letters that correspond to the MBTI indicators (e.g. I, N, T, J) or combinations of them (e.g. INTJ). Because of this, we generated an additional version of the MBTI dataset, where each possible MBTI indicator and their combinations were masked, replacing them with a special

symbol used to denote the unknown word (for a detailed list, check A1 in S1 File). This masking process affected a small portion of the words of the dataset, more specifically, a 1.6% of them. However, in order to verify that any possible change in the results when masking the words was exclusively due to masking the MBTI words and not because of the quantity of masked words, we decided to create a third version of the MBTI dataset where we randomly masked the same proportion of words as for the masked dataset. We will refer to this version as the randomly masked MBTI dataset.

Finally, we preprocessed the text of the Essays and MBTI datasets (including original and masked versions) using a standard preprocessing library (https://github.com/s/preprocessor), which incorporates basic cleaning and filtering operations.

By leveraging these diverse datasets, we aim to explore the nuances of personality representation in natural language despite the inherent challenges and limitations posed by each dataset.

## Analytic techniques

1. **Log-Odds Ratio with Informative Dirichlet Prior (IDP)** [58]: by using the log-odds ratio method we seek to identify words that significantly differ in usage between the positive and negative cases of a certain trait or personality type. In this context, the odds of a particular word is the ratio of the probability of the word not occurring in a set of texts to the probability of it not occurring. When calculating the log-odds ratio for a word, we obtain the logarithm of dividing the odds for a word when a trait or personality type is present by the odds when it is not. In our work, we calculate the log-odds ratio using the IDP method.

2. **Language Models**: a language model is a probabilistic model of a natural language. We used BERT (Bidirectional Encoder Representations from Transformers) [33] and its optimized version RoBERTa (a Robustly Optimized BERT Pretraining Approach) [50]. They are notable for their huge improvement with respect to previous state-of-the-art language models. Essentially, these models process text through a series of interconnected components. First, text is broken into tokens, which are the basic units of meaning in the input, in a process called tokenization. These tokens can be words, but often subwords are used—parts of words that constitute common prefixes, suffixes, or root words— to handle a wider vocabulary more efficiently. Next, the tokens are converted into numerical representations called embeddings, which incorporate semantic and syntactic information. When generating the embeddings, a bidirectional approach is adopted, considering both left and right contexts for each token. The heart of BERT and RoBERTa consists of multiple attention layers, each containing multi-head attention mechanisms and feed-forward networks. These layers progressively refine the initial token representations, capturing complex contextual relationships. Both BERT and RoBERTa are pre-trained with a large collection of texts composed of millions of words, resulting in models with a deep understanding of language structure and meaning. The pre-trained models can be adapted to perform different tasks by means of a fine-tuning process, which typically requires much less training data than model pre-training. In our work, the fine-tuning process is used to adapt BERT and RoBERTa for predicting personality traits or typologies for the texts contained in the Essays and MBTI datasets. BERT and RoBERTa were used in this study because of their extensively demonstrated performance in multiple language processing tasks, including text classification, and also because of their public availability, that ensures the reproducibility of results. We did not consider the use of the DeBERTa model [42], which can be seen as an enhanced

version of RoBERTa, because of its substantially higher memory requirements with respect to BERT and RoBERTa. Such requirements made it very difficult to process the long texts contained in the Essays and MBTI datasets using available hardware without introducing aggressive text truncation or fragmentation steps.

3. **Integrated Gradients** [84]: a method that helps us understand how a model makes decisions in a classification task. In the context of our work, the technique will allow us to identify which words are important for the BERT and RoBERTa models when making classification predictions. More specifically, given a prediction, the technique returns an attribution score for each word, that can be positive or negative. If a particular word has a positive attribution score, this means that the word increased the likelihood of the prediction returned by the classifier. For instance, assuming that the classifier has predicted the trait Agreeableness for a text belonging to the Essays dataset, the word "forgive" having a positive attribution score means that the presence of such a word in the text increased the likelihood of the text being classified as Agreeableness. In contrast, a negative attribution score means that the word decreased the likelihood of the prediction. The attribution scores of individual words in a text can be combined to provide an overall score for the entire text. In the context of our work, integrated gradients constitute a particularly interesting technique, since it enables a more detailed analysis of the words relevant to a particular personality trait or type. More specifically, the words can be studied in individual texts instead of in groups of texts, as it is the case of less sophisticated techniques such as the log-odds ratio with IDP explained above. We chose to use integrated gradients as an explainability technique due to its desirable properties: (i) they constitute an approximation of the Aumann-Shapley values, see [36], which are axiomatically justified (axioms refer to conditions that the explainability technique should ideally satisfy), and (ii) they can be efficiently calculated as long as the model used is piecewise differentiable.

4. **Word Clouds** [40]: word clouds constitute a very useful tool to summarize textual data. Word clouds visually display words, typically sized according to their frequency in a set of texts. In our work, word frequency is replaced by a measure of word importance according to the attribution scores generated by the integrated gradients technique. Due to the fact that such a technique produces one attribution score per each word contained in a text, it is necessary to define summarization techniques that produce a single score per each word (see next section for more details). Generating word clouds with summarized attribution scores will allow us to study significant words for the different personality traits or types, and check for congruence with the theory supporting the Big Five traits and the MBTI typologies, respectively.

## Evaluation metrics

When calculating the IDP-based log-odds ratios for the different words and personality traits or types, we obtain the z-score to provide a measure of statistical significance. This measure is useful for hypothesis testing to determine whether the observed effect of a particular word is likely to be genuine or due to random chance. A large, positive z-score for a word indicates its strong relevance to a particular personality trait or type. Conversely, a small, negative z-score suggests that the word is more likely associated with the absence of that trait or type.

The performance of the classification using BERT and RoBERTa was evaluated with two metrics: accuracy and Area Under the Curve (AUC). Accuracy is defined as the ratio of the number of correct predictions to the total number of predictions made. It measures how well the classifier correctly identifies both positive and negative instances in a dataset. On the

other hand, AUC refers to the area under the Receiver Operating Characteristic (ROC) curve. The ROC curve is a graphical representation that illustrates the diagnostic ability of a binary classifier system as its discrimination threshold is varied. The AUC measures the model's ability to distinguish between classes.

Both performance metrics go from 0 to 1 with values approaching 1 showing better performance. In two-class classification problems where the data is imbalanced (one of the classes is more frequent than the other), the accuracy can be misleading, since the classifier can learn to always return the more frequent class as output, resulting in a poor model with a high accuracy (e.g. if 95% of instances belong to one class, a model that always predicts the majority class will have 95% accuracy). In contrast, the AUC is robust to class imbalance.

While classification metrics provide an overall assessment of model performance, they don't offer insights into which specific words or features contribute most to the model's decisions. To address this, we employed several techniques on the attribution scores obtained by the integrated gradients technique to measure word importance, particularly in the context of generating word clouds and visualizing influential words.

To effectively summarize and interpret these attribution scores, we developed multiple summarization techniques. These techniques allow us to quantify and visualize the importance of individual words in both positive and negative attribution to the model's decisions. Specifically, we evaluated word importance using the following methods:

- **Frequency counting**: we selected words based on their rate of appearance in either the positive and the negative attribution score. This highlights words that consistently contribute to or detract from a classification.
- **Accumulation**: we selected words based on their attribution scores and accumulated the score for each time the word appeared. This emphasizes words that consistently contribute to the model's decision across multiple instances.
- **Maximum attribution**: we selected only the highest score for each word. This captures words with a strong impact in at least one case.
- **Average**: where we selected the accumulation of the word and divided by their frequency counting. This balances frequency and impact, providing a measure of a word's overall importance across all instances.
- **Geometric mean**: we calculated the geometric mean of the accumulated and maximum scores. This technique gives weight to both consistent performance (accumulation) and peak performance (maximum), offering a balanced view of a word's importance.

Each technique provides a different perspective on the data, allowing us to capture various aspects of the attributions. For each summarization technique, word clouds were generated to visually represent the significant words contributing to both positive and negative attributions for each label. We also created bar plots for the average and geometric mean techniques, as these better capture overall tendencies and influential examples (see OSF for all the bar plots). However, these visualization techniques fall short when representing entire essays or posts.

## Experiments configuration

The Log-odds ratio with IDP method was implemented by means of a publicly available software package (https://github.com/kornosk/log-odds-ratio), which was applied to both the Essays and MBTI datasets. For the case of the MBTI dataset, the IDP-based log-odds ratio technique was applied to the original and masked versions. Regarding the language models, before applying BERT and RoBERTa for trait/type identification, it was necessary to fine-tune the parameters of both models. For this purpose, we first split each dataset with a 90:10

training-test ratio respectively. Additionally, we reserved a random 10% of training data to create the validation set for all models. Once the partitions for each dataset were created, we used a grid search procedure to find the best model. The search explored different values of the hidden and attention dropout probabilities (with the aim to reduce model overfitting) and also of the learning rate (so as to control how quickly or slowly the model learns). For the Essays dataset, 5 training epochs were executed, while for the MBTI dataset, only 3 were executed, due to the greater computational requirements, and also because of the fact that overfitting often started before the third epoch. Due to its robustness against class imbalance, the AUC metric was used to evaluate model performance. In all cases, we kept the model achieving a higher AUC for the validation set from all the epochs.

The fine-tuning process was executed using the "bert-base-uncased" and "roberta-base" versions of BERT and RoBERTa, respectively. Both versions were operated by means of the Huggingface Transformer library [91] for PyTorch [63].

From the traits, Agreeableness was the only one where RoBERTa was used to implement a classifier, given that BERT did not produce good performance results. In contrast, the MBTI dataset was only processed with BERT, due to the fact that both, BERT and RoBERTa, achieved very similar performance, but the memory requirements involved in the calculation of the attribution scores were significantly higher for RoBERTa, due to the fact that its tokenizer generated substantially longer tokenized texts (BERT and RoBERTa follow different tokenization strategies). Additionally, for MBTI we focused mainly on the AUC metric, given the disproportion in the frequency of most of the classes in the dataset (Table 4 in S1 File & Fig 1). Finally, it should be stressed out that model training was carried out for original, masked and randomly masked MBTI datasets.

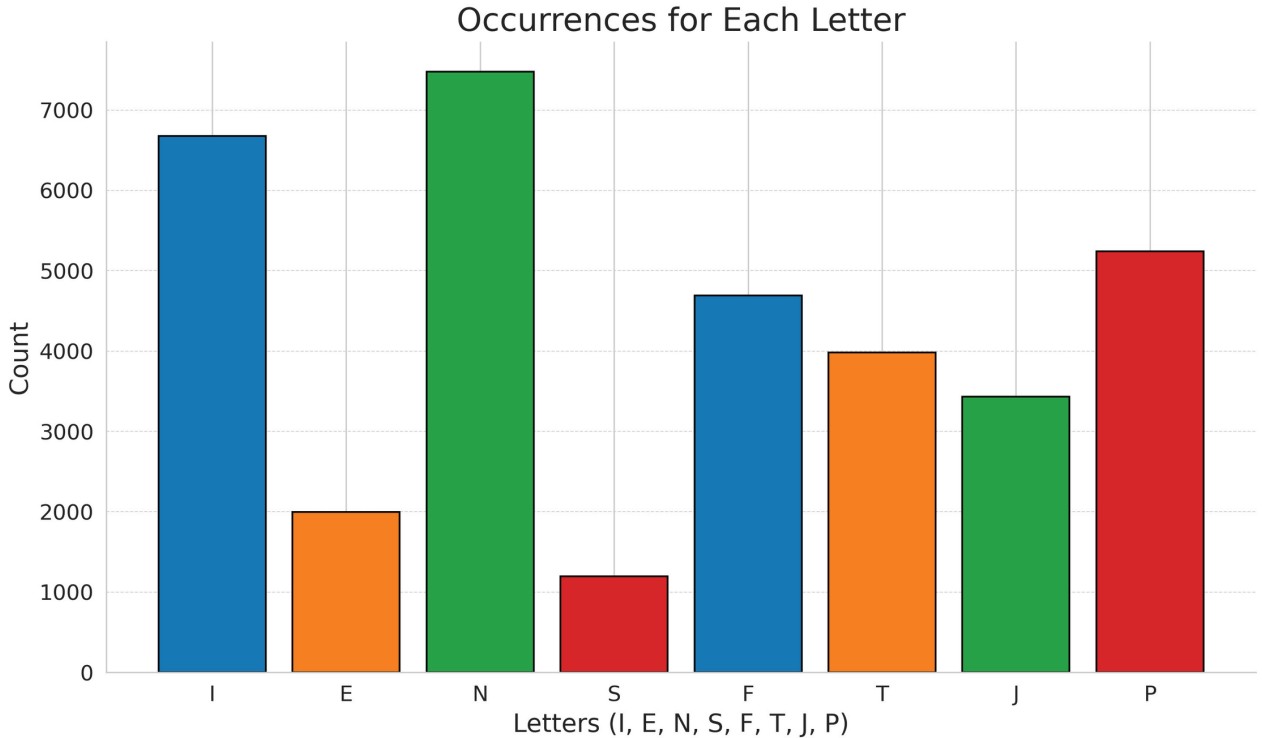

**Fig 1. Occurrences for each letter of the MBTI in the personality café dataset.** Each letter is distributed unequally in the dataset.

On the other hand, we used the Transformers Interpreter library [66], which provides an integrated gradients implementation for transformers, to compute the attribution scores at the word and whole text levels for each dataset.

All the experiments were executed in a computer with an Intel Core I7 processor and 32GB of RAM. The BERT and RoBERTa models, including fine-tuning and integrated gradients computation, were operated by means of an NVIDIA GeForce RTX 3090 GPU.

## Data analysis procedure

The Essays dataset was used to fine-tune five models, four of them employing bert-base (O, C, E, & N) and one employing roberta-base (A). On the other hand, the MBTI dataset was used to fine-tune eight models with bert-base: four with the original data and four with the data masked.

After the models were trained, we used the Transformers Interpret library to attribute the models' prediction to individual words in the text. This process assigns two types of attribution scores: (1) word-level scores: indicating the importance of each word in contributing to the model's prediction; and (2) text-level scores: indicating the overall importance of a text for a given prediction.

To focus on significant results and ensure correct interpretation of attribution scores, we implemented several criteria. We verified congruence between true and predicted labels from the classification task, meaning we only considered instances where the model's prediction matched the actual label. For example, a high attribution score for a word in relation to the trait of agreeableness is meaningful only if the corresponding class prediction is also agreeableness. In our code and plots, this relationship is indicated by label 1 (classified as presenting the trait of interest) and label 0 (classified as not presenting the trait of interest).

Additionally, we only considered words that appeared at least 10 times in the texts highlighted by the Transformer Interpret. This approach allowed us to identify the most relevant and frequently occurring words in correctly classified instances. We also analyzed the contexts where entire paragraphs containing high-attribution words achieved the highest scores. This approach helps us see how the model interprets the broader context in which each class appears.

We identified the top 5 highest word-level attribution scores from the average and geometric mean bar plots for both label 1 and label 0, for each trait and MBTI dimension. We then selected texts where these top words appeared, focusing on those texts that also had high text-level attribution scores.

We read these selected texts to understand the context of the highly attributed words and to evaluate the performance of the classification. For the sake of completeness, we also read the texts with the highest overall text-level attribution scores to ensure we captured the most representative examples for each class (having the trait/type or not). Each output was read and compared with the content of the personality items used by the original authors of the data to see if its content matched aspects of the theory of personality traits and types.

In summary, to filter the most significant and pertinent texts, we performed the visualization on each of the classification classes (label 1 or 0) with three different approaches: (1) we obtained the 10 texts with the highest overall text-level attribution scores for each class, (2) we identified the 5 words with the highest attribution scores using the geometric mean summarization technique, then selected the 10 texts containing these words that had the highest text-level attribution scores, and (3) we followed the same process as in (2), but using the average summarization technique to identify the highest-attributed words. This multi-faceted

approach allowed us to capture texts that were significant both at the overall text level and for their highly impactful individual words.

For the MBTI dataset, we explored our hypothesis by masking the terms in the data that represented all the types from the MBTI list and other elements that might resemble the MBTI list. This list was obtained through a process of trial and error. In a similar way that occlusion techniques work [95] we decided to check the improvement of performance metrics to determine the weight these terms had in the model accuracy (for a detailed list of these terms check A1 in S1 File).

## Results

Our analysis yielded several key findings regarding the relationship between the manifestation of personality on language, uncovering previously unexamined patterns in how personality traits emerge in natural language use. We present these results in three main parts: first, we examine the words most relevant to each personality trait or type as identified by the Log-Odds Ratio with Informative Dirichlet Prior method. Next, we report the classification performance metrics for our models. Finally, we delve into the insights gained from applying explainability techniques to our classification results, revealing novel patterns in how language models process and interpret personality-relevant information. These findings collectively shed light on the linguistic markers of personality and the efficacy of NLP techniques for personality assessment, while providing unprecedented insight into how these models identify personality traits in text.

### Log-Odds Ratio with informative Dirichlet Prior

The goal of this technique is to identify those words that are relevant for the different personality traits or typologies. Tables 1, 2 and 3 present the top words identified for each trait in the Essays dataset, or for the personality types in the original and masked versions of the MBTI dataset, respectively.

The Big Five analysis reveals a mix of words that can be associated with their respective traits, as well as some that lack an evident conceptual link. Table 1 presents the top words and their z-scores for each of the Big Five traits.

For Openness to Experience, word related to art and abstract concepts appear with positive z-scores, such as 'guitar' (z-score = 1.61) and 'music' (z-score = 1.34), as well as 'world' (z-score = 1.4) and 'words' (z-score = 1.31). Conversely, words associated with routine and concrete experiences show negative z-scores, including 'home' (z-score = –2.6), 'school' (z-score = –2.1), or 'class' (z-score = –2).

Conscientiousness displays a less clear pattern. Words like 'hope' (z-score = 1.3), '1' (z-score = 1.25), 'tonight' (z-score = 1.24), and 'able' (z-score = 1.2) show positive z-scores, while 'im' (z-score = –1.73), 'Damn' (z-score = –1.21), and 'Squirrel' (z-score = –1.2) have negative z-scores. The connection between these words and the trait's conceptualization is not immediately apparent.

For Extraversion, words related to social activities appear with positive z-scores, like 'Sorority' (z-score = 1.8), 'Fun' (z-score = 1.53), 'Boyfriend' (z-score = 1.42), and 'Love' (z-score = 1.32). On the other hand, ambiguous words appear with negative z-scores, like 'Perhaps' (z-score = –1.62), 'Da' (z-score = –1.4), 'Cold' (z-score = –1.03), and 'Write' (z-score = –1.03).

**Table 1. Z-scores when calculating Log-Odds ratio with informative Dirichlet Prior for the essays dataset.**

| Trait | Word | Positive z score | Word | Negative z score |
|---|---|---|---|---|
| Openness to Experience | Guitar | 1.61 | Home | −2.6 |
| | World | 1.4 | Go | −2.2 |
| | Cat | 1.35 | Going | −2.14 |
| | Music | 1.34 | School | −2.1 |
| | Words | 1.31 | Class | −2 |
| Conscientiousness | Hope | 1.3 | im | −1.73 |
| | 1[1] | 1.25 | Damn | −1.21 |
| | Tonight | 1.24 | Squirrel | −1.2 |
| | Able | 1.2 | Bla | −1.14 |
| | Week | 1.2 | Finger | −1.11 |
| Extraversion | Sorority | 1.8 | Perhaps | −1.62 |
| | im | 1.62 | Da | −1.4 |
| | Fun | 1.53 | 1[1] | −1.31 |
| | Boyfriend | 1.42 | Cold | −1.03 |
| | Love | 1.32 | Write | −1.03 |
| Agreeableness | Family | 1.5 | Fucking | −1.5 |
| | Really | 1.2 | Stupid | −1.4 |
| | Weekend | 1.1 | 1[1] | −1.34 |
| | Home | 1.1 | Read | −1.22 |
| | Emily | 1.05 | Bla | −1.2 |
| Neuroticism | Feel | 1.55 | Beat | −1.22 |
| | Want | 1.41 | Pledge | −1.2 |
| | Scared | 1.4 | Game | −1.12 |
| | Stressed | 1.35 | Thats | −1.03 |
| | Boyfriend | 1.31 | Coach | −1.02 |

Agreeableness shows words related to relationships with positive z-scores, like 'Family' (z-score = 1.5), 'Home' (z-score = 1.1), and 'Emily' (z-score = 1.05). Conversely, profane vocabulary obtained negative z-scores, like in 'Fucking' (z-scores = −1.5) and 'Stupid' (z-scores = −1.4).

Neuroticism obtained words related to affective states like 'Feel' (z-scores = 1.55), 'Want' (z-scores = 1.41), 'Scared' (z-scores = 1.4), with positive z-scores; and words with no clear relationship with negative z-scores, like 'Beat' (z-scores = −1.22), 'Pledge' (z-scores = −1.2), 'Game' (z-scores = −1.12).

Regarding the MBTI, our H4 is supported by the IDP-based log-odds ratio results, which predominantly show words syntactically related to the typologies. The classification's unreliability on this dataset is evident in how frequently individuals discuss their own type. As shown in Table 2, the leading words for each dimension are consistently the letter abbreviations of the types themselves. For the I/E dimension, the words 'INFP' (z-score = 5.12), 'INFJ' (z-score = 3.6), 'INTP' (z-score = 3.33) are the top words indicating Introversion, while 'ENFP' (z-score = −13.8), 'ENTP' (z-score = −13.3), 'ENTPs' (z-score = −8) are the most indicative of Extraversion.

This pattern is consistent across all MBTI dimensions. For the N/S dimension, 'INFJ' and 'INFP' are top indicators for Intuition, while 'ISTP' and 'ISFP' lead for Sensing. In the F/T dimension, 'INFP' and 'INFJ' are strong indicators for Feeling, with 'INTP' and 'INTJ' leading for Thinking. Finally, for the J/P dimension, 'INFJ' and 'INTJ' are top indicators for Judging, while 'INFP' and 'INTP' lead for Perception (see Table 2). These results strongly support H4, demonstrating that the most influential elements include a high frequency of MBTI categories.

After applying the masking, the types no longer appeared among the predictive words. Table 3 presents these new words and their z-scores. The words used obtain scores notably

**Table 2. Z-scores when calculating Log-Odds ratio with informative Dirichlet Prior for the MBTI—full dataset.**

| Dimensions | Word | Positive z score (I N F J) | Word | Negative z score (E S T P) |
|---|---|---|---|---|
| Introversion -<br>Extraversion | INFP | 5.12 | ENFP | −13.8 |
| | INFJ | 3.6 | ENTP | −13.3 |
| | INTP | 3.33 | ENTPs | −8 |
| | INFPs | 2.72 | ENTJ | −7.7 |
| | INFJs | 2 | ENFPs | −7.53 |
| | INTPs | 1.92 | ENFJ | −6.02 |
| | music | 1.8 | ESTP | −4.92 |
| | dream | 1.72 | 7w6 | −4 |
| | games | 1.66 | ne | −4 |
| | family | 1.6 | ENTJs | −4 |
| Intuition - Sensing | INFJ | 3.34 | ISTP | −9.32 |
| | INFP | 3.3 | ISFP | −7.8 |
| | INTJ | 3.13 | ISFJ | −7 |
| | INFPs | 2.56 | ISTJ | −6.6 |
| | INFJs | 2.35 | ESTP | −4.7 |
| | â | 2.25 | ISTPs | −4.7 |
| | INTP | 2.2 | ISFJs | −4 |
| | INTJs | 2.05 | ISFPs | −3.7 |
| | ENTP | 1.94 | ESFJ | −3.6 |
| | ENFP | 1.82 | rave | −3.45 |
| Feeling - Thinking | INFP | 13.32 | INTP | −12.7 |
| | INFJ | 11.32 | INTJ | −10.8 |
| | Feel | 9.53 | ENTP | −9.14 |
| | :) | 9.41 | INTPs | −7.42 |
| | Love | 8.97 | INTJs | −6.1 |
| | INFPs | 7.96 | ENTJ | −5.31 |
| | ENFP | 7.3 | ENTPs | −5.2 |
| | INFJs | 5.57 | ISTP | −4.89 |
| | ENFJ | 5.45 | ti | −4.2 |
| | really | 5.22 | shit | −3.9 |
| Judging - Perception | INFJ | 14.1 | INFP | −8.83 |
| | INTJ | 10.6 | INTP | −7.4 |
| | INFJs | 8.14 | INFPs | −5.45 |
| | INTJs | 6.5 | ENTP | −5.43 |
| | ni | 6.14 | ENFP | −5.3 |
| | ENFJ | 4 | INTPs | −4.21 |
| | ISFJ | 4 | ISTP | −4.01 |
| | ISTJ | 3.28 | ´ | −3 |
| | ENTJ | 3.3 | ISFP | −3 |
| | dear | 2.7 | ENTPs | −2.8 |

below the ones reported with no masking. Additionally, in some of the types, the term '[UNK]', which for the BERT model, represents the unknown word that was used to mask words, appears with a high z-score suggesting that the masked words had an influential effect on the data. Importantly, this term appears codified as 'unk' with the Logs-Odds Ratio with Informative Prior.

For the I/E dimension, 'music' (z-score = 1.78) and 'dream' (z-score = 1.72) are the top words indicating Introversion, while 'unk' (z-score = −9.31) and '7w6' (z-score = −3.98) are the most indicative of Extraversion. For the N/S dimension, 'â' and 'â' lead for Intuition whereas 'rave' and 'rant' lead for Sensing. For the F/T dimension, 'feel' and ':)' are the top words for Feeling, and 'ti' and 'shit' for Thinking. Finally, for the J/P dimension 'ni' and 'dear' lead for Judging and ' '' ' and 'really' for Perception.

**Table 3. Z-scores when calculating Log-Odds ratio with informative Dirichlet Prior for the MBTI—masked dataset.**

| Dimensions | Word | Positive z score (I N F J) | Word | Negative z score (E S T P) |
|---|---|---|---|---|
| Introversion - Extraversion | music | 1.78 | unk | −9.31 |
| | dream | 1.72 | 7w6 | −3.98 |
| | games | 1.66 | ne | −3.95 |
| | family | 1.59 | 7w8 | −3.8 |
| | feel | 1.58 | sx | −3.71 |
| | world | 1.55 | lol | −3.51 |
| | death | 1.52 | :d | −3.5 |
| | years | 1.46 | xd | −3.11 |
| | feeling | 1.42 | guys | −3.08 |
| | quiet | 1.4 | 8w7 | −3 |
| Intuition - Sensing | â | 2.78 | rave | −3.45 |
| | ã | 2.73 | rant | −3.01 |
| | , | 1.88 | unk | −2.8 |
| | f | 1.77 | si | −2.46 |
| | world | 1.74 | se | −2.2 |
| | ... | 1.67 | 360v | −2.15 |
| | human | 1.33 | type | −2.1 |
| | [blank space] | 1.19 | niss | −1.81 |
| | Œ | 1.15 | digger | −1.8 |
| | universe | 1.11 | posh | −1.75 |
| Feeling - Thinking | feel | 9.54 | ti | −4.15 |
| | :) | 9.41 | shit | −3.86 |
| | love | 8.98 | use | −3.64 |
| | really | 5.23 | knowledge | −3.61 |
| | :d | 5.01 | argument | −3.39 |
| | happy | 4.96 | nt | −3.3 |
| | feeling | 4.91 | intelligence | −3.19 |
| | thank | 4.9 | logic | −3.17 |
| | beautiful | 4.6 | point | −3.16 |
| | ... | 4.29 | physics | −3.13 |
| Judging - Perception | ni | 6.16 | ´ | −2.98 |
| | dear | 2.71 | really | −2.63 |
| | others | 2.66 | ne | −2.47 |
| | fe | 2.51 | shit | −2.41 |
| | 1/2 | 2.44 | music | −2.39 |
| | rave | 2.29 | fuck | −2.3 |
| | ni - ti | 2.1 | yeah | −2.26 |
| | rant | 2.09 | pretty | −2.16 |
| | welcome | 2.06 | like | −2.11 |
| | intuition | 2.05 | :d | −2.1 |

Notably, terms like '7w6' or '7w8' also appear in this analysis. These terms belong to the Enneagram, a different and similarly problematic theory of personality [45]. Their presence indicates the broad knowledge of personality theories among Personality Café participants. Despite these limitations, some influential words show conceptual coherence with the MBTI theory. In the case of F/T, the words align with theoretical expectations. However, this alignment might still be influenced by the self-reference effect we observed with the type abbreviations, as participants may use language similar to the type names when discussing their personality.

## Accuracies and AUCs

The Essays dataset followed a data partition procedure that included training, validating, and testing to avoid overfitting. The accuracies for the Big Five obtained in the test dataset were 0.637 for Openness to Experience, 0.601 for Conscientiousness, 0.620 for Extraversion, 0.590 for Agreeableness, and 0.620 for Neuroticism (see Table 4). The values between the validation

**Table 4. Accuracies and AUCs for the Big Five.**

| Trait | Validation dataset | | Test dataset | |
|---|---|---|---|---|
| | Accuracy | AUC | Accuracy | AUC |
| Openness to Experience | 0.649 | 0.651 | 0.637 | 0.640 |
| Conscientiousness | 0.614 | 0.614 | 0.602 | 0.599 |
| Extraversion | 0.627 | 0.629 | 0.620 | 0.620 |
| Agreeableness | 0.636 | 0.633 | 0.590 | 0.590 |
| Neuroticism | 0.614 | 0.614 | 0.620 | 0.620 |

and the testing dataset do not differ much, indicating no overfitting. The accuracies obtained for the Big Five support H1, indicating moderate classification accuracy.

Comparing our results (Table 4) with previous studies (Table 1 in S1 File) shows that our accuracies are comparable to most of the accuracies reported in prior studies, with Ramezani, Feizi-Derakhshi, and Balafar [69] being the only exception. Their higher accuracy likely stems from their use of a Knowledge Graph Attention Network (KGrAt-Net) rather than BERT, representing text as knowledge graphs and applying graph attention mechanisms unlike our contextualized approach. BERT and RoBERTa are specialized in language processing, trained on massive text data, and publicly available, ensuring reproducibility. In contrast, the KGrAt-Net is not public and its implementation is not straightforward. While not our primary objective, our accuracy results lend credibility to the generalizability of our conclusions on explainability that will be presented later.

The accuracy and the AUC for the original, masked, and randomly masked versions of the MBTI dataset are presented in Table 5. For the three versions, important discrepancies can be observed between the accuracy and AUC metrics, being the former consistently higher. This is due to the fact that the accuracy metric is not robust against the class imbalance that is present in the MBTI dataset (Table 4 in S1 File & Fig 1). Additionally, it is misleading to compare the MBTI accuracy with that of the Big Five dataset, since the latter does not present the class imbalance problem. In summary, these observations support H2. On the other hand, it is striking to observe the large difference between the AUC results for the original and masked versions of MBTI, revealing the true performance of the model when the self-reference MBTI terms are removed from the texts being analyzed.

The I/E original AUC was 0.739 while the masked AUC was 0.596; interestingly, the randomly masked AUC remained at 0.729. For N/S, the original AUC was 0.735, the masked AUC decreased to 0.565, while the randomly masked AUC slightly increased to 0.751 (although we attribute the small increase to chance). The F/T dimension showed the least variation, with an AUC of 0.816 and 0.814 for the original and randomly masked datasets respectively, and 0.744, for the masked dataset. Finally, for J/P, the original AUC was 0.743, the masked AUC decreased to 0.600, and the randomly masked AUC maintained a similar

**Table 5. Accuracies and AUCs for the MBTI with masked and original types.**

| Types pair | Original | | Masked | | Randomly Masked | |
|---|---|---|---|---|---|---|
| | Accuracy | AUC | Accuracy | AUC | Accuracy | AUC |
| I/E | 0.834 | 0.739 | 0.764 | 0.596 | 0.837 | 0.729 |
| N/S | 0.888 | 0.735 | 0.862 | 0.565 | 0.878 | 0.751 |
| F/T | 0.818 | 0.816 | 0.748 | 0.744 | 0.815 | 0.814 |
| J/P | 0.757 | 0.743 | 0.631 | 0.600 | 0.750 | 0.731 |

level at 0.731. Notably, AUC was nearly identical between original and randomly masked datasets, in clear contrast with the masked dataset, showing the stark influence of the MBTI terms on the classification.

These results show strong support for H4, namely, knowing the MBTI terms previously assigned to oneself will impact the classification performance. Notably, the F/T dimension was the one with less difference in AUCs between original and masked datasets, suggesting it was less susceptible to the bias the others suffered from.

Our results support the use of BERT or RoBERTa for personality prediction from human-generated text, but with a crucial caveat: the effectiveness depends heavily on the quality of the training data. While these models performed well with the Essays dataset, their performance on the original MBTI dataset highlights the importance of using truly informative data, free from self-reference biases, for accurate personality assessment.

### Explainability—word attribution scores

To analyze the importance of individual words in the model's decision-making process, we employed several summarization techniques for the word attributions. While each technique offers a unique perspective on word importance, we chose to focus primarily on the geometric mean for our analysis and visualization. We present barplots of the top words with the highest attribution scores for Agreeableness, the original Feeling/Thinking, and the masked Feeling/Thinking, obtaining a rank of the most influential words (Figs 2, 3, and 4). These bar

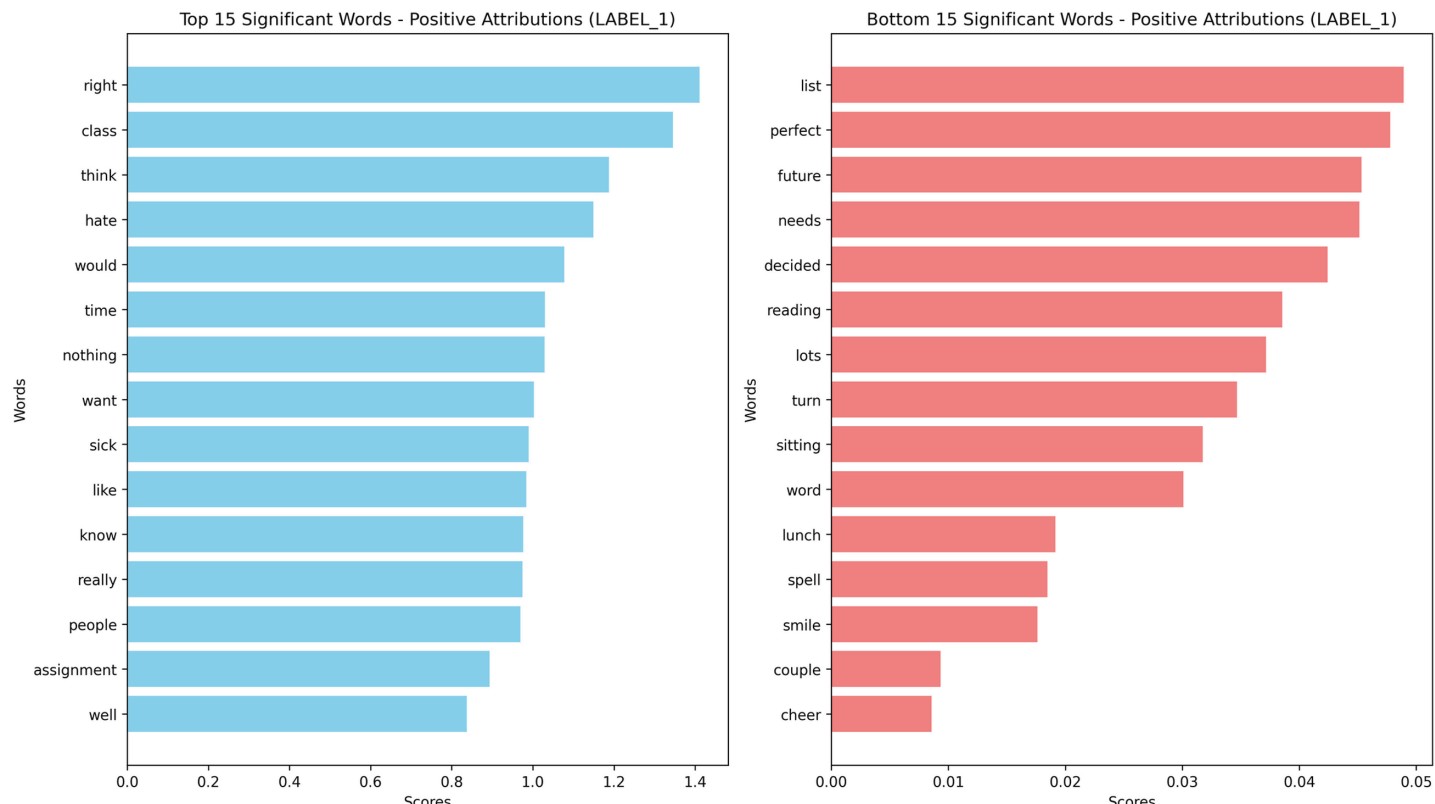

**Fig 2. Bar plot for the geometric mean positive attribution scores for Agreeableness.** Visualization of the most important words for Agreeableness.

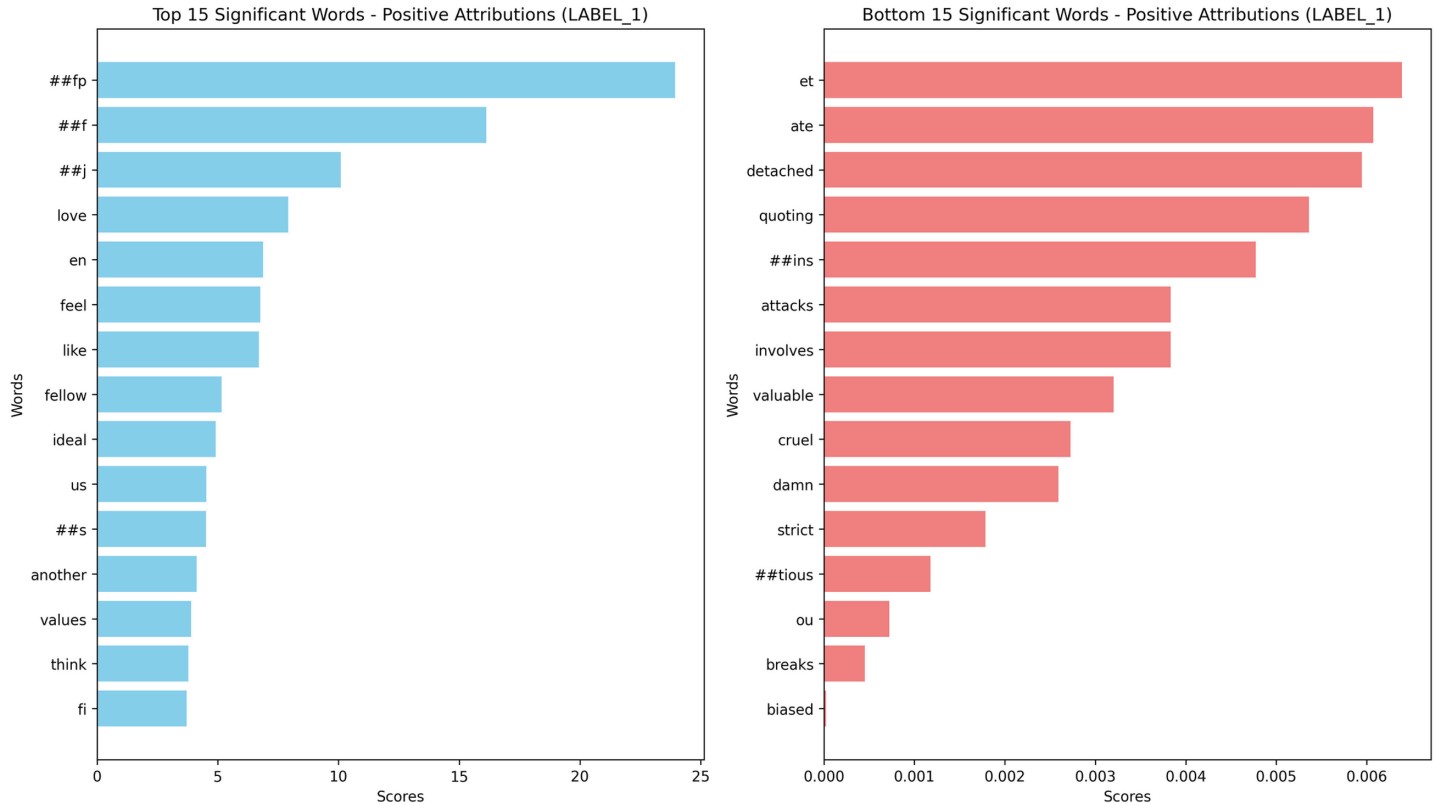

**Fig 3. Bar plot for the geometric mean positive attribution scores for Feeling/Thinking original dataset.** Visualization of the most important words for Feeling/Thinking.

plots provide a clear and concise visualization of the words that have the highest impact on the model's prediction for each trait.

To complement this quantitative analysis, we also generated word clouds (Figs 5, 6 and 7) visualizing the geometric mean scores for each trait. Below, we analyze these results for each of the Big Five personality traits. The complete list of barplots and word clouds are available in the OSF (available in this repository).

- **Agreeableness** (Figs 2 & 5): The top contributors include words like 'right', 'class', and 'think', which may not immediately align with the conceptualization of the trait. However, words such as 'like' and 'well' also appear, more closely aligning with the trait's features. The presence of 'hate' among the top contributions highlights the complex nature of language use in relation to personality traits.
- **Conscientiousness**: Words like 'work', 'homework', 'assignment', and 'classes' align with the trait. Interestingly, 'people' and 'love' are two top contributors possibly reflecting the responsibility entailed in social relationships. The presence of 'money' and 'time' might reflect financial and time organization.
- **Extraversion**: 'Sorority' and 'college' top the list, strongly aligning with the social aspect of the trait. Words like 'awesome', 'like', and 'definitely' may reflect the enthusiastic expression often associated with this trait. Social and active words are represented in 'football', 'weekend', and 'tonight'.

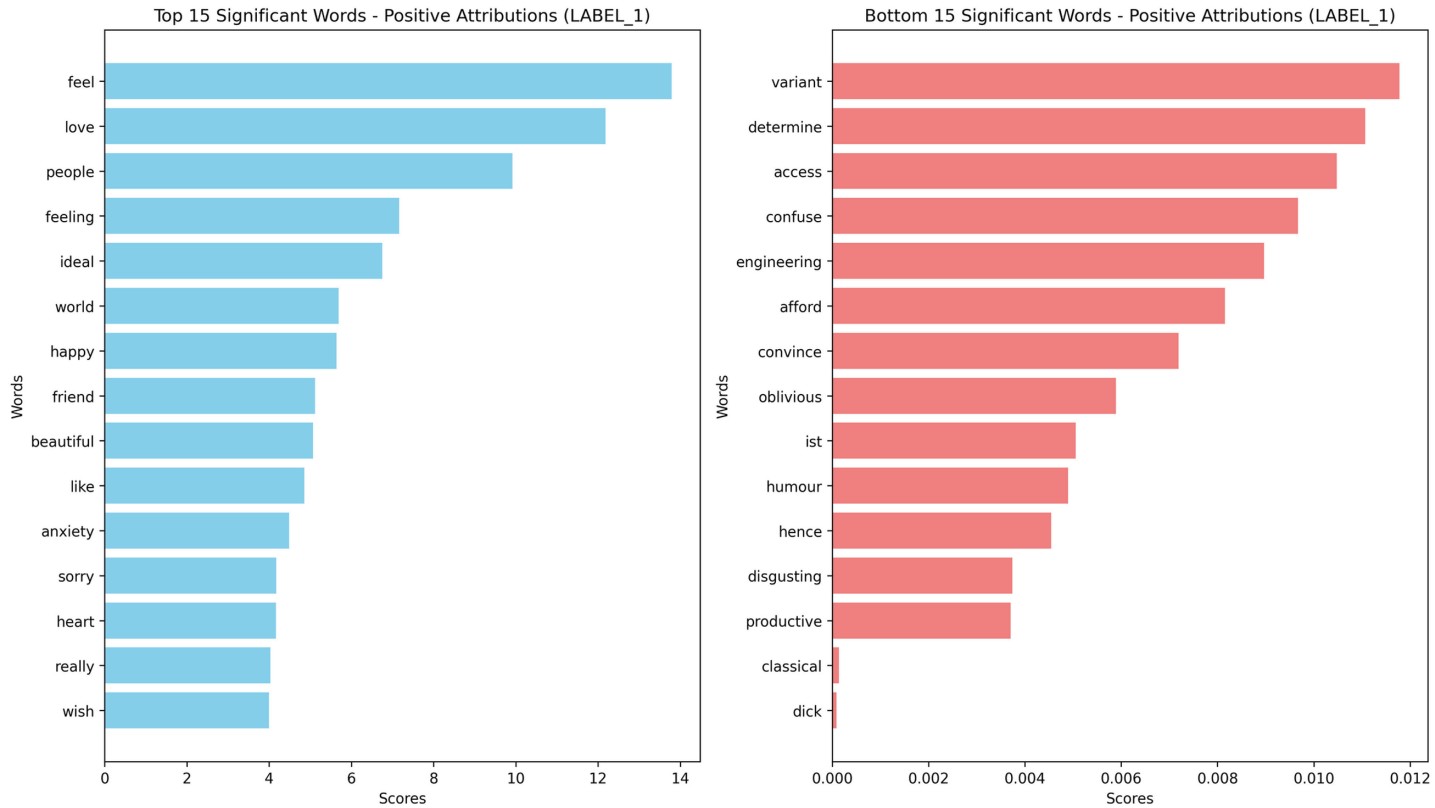

**Fig 4. Bar plot for the geometric mean positive attribution scores for Feeling/Thinking masked dataset.** Visualization of the most important words for Feeling/Thinking.

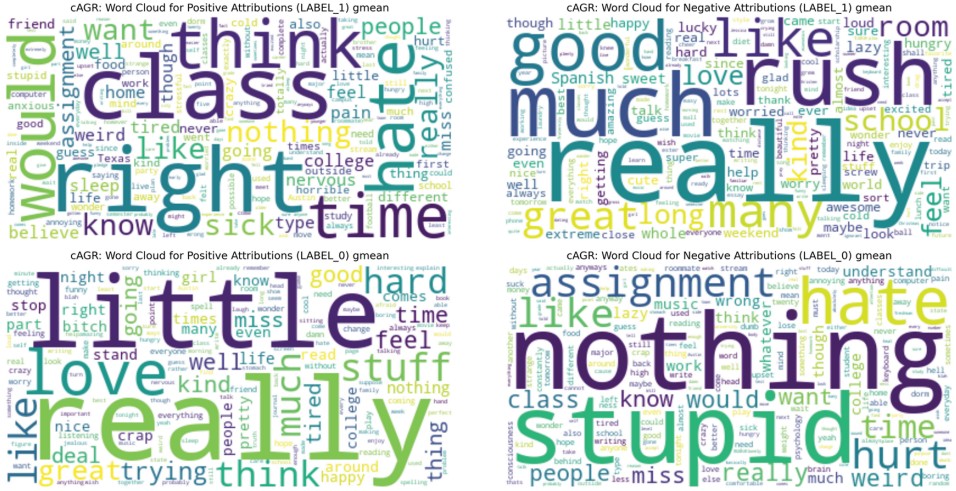

**Fig 5. Word Cloud for the geometric mean positive attribution scores for Agreeableness.** Overview of the most important words for Agreeableness.

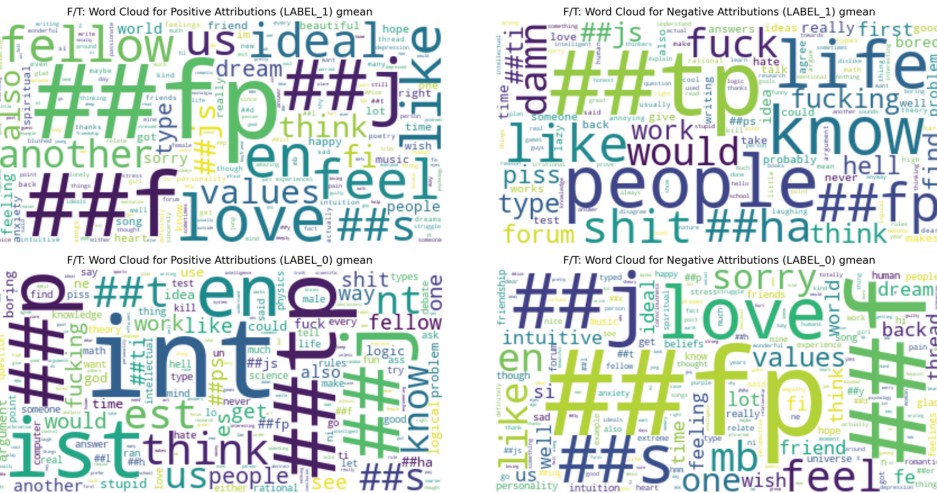

**Fig 6. Word Cloud for the geometric mean positive attribution scores for Feeling/Thinking original dataset.**
Overview of the most important words for Feeling/Thinking.

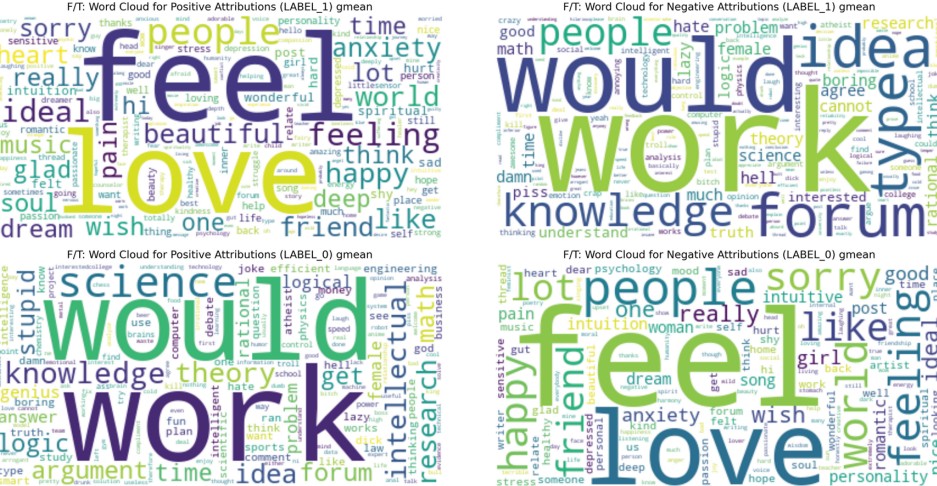

**Fig 7. Word Cloud for the geometric mean positive attribution scores for Feeling/Thinking masked dataset.**
Overview of the most important words for Feeling/Thinking.

- **Neuroticism**: 'Hate' tops the list, aligning with the negative emotionality of this trait. Words like 'hurt' and 'wrong' reinforce the anxious and worrying aspects of this trait. Interestingly, 'love' and 'people' also appear high on the list, possibly explaining intense emotional experiences or the moderate accuracy in the classification.
- **Openness to Experience**: 'College' and 'class' are among the top five on the list, potentially reflecting intellectual curiosity. 'Music' also appears, aligning with aesthetic interests. Words like 'life', 'time', 'love', and 'people' also appear indicating a broad interest in abstract concepts.

These findings reveal the complex nature of how personality traits manifest in language use. While some words clearly align with our understanding of each trait, others suggest more nuanced or context-specific expressions. The prevalence of college life-related words across traits is an indicator of the demographics of the participants.

The MBTI barplots from the masked dataset showed some reference to the MBTI's theory but the results are less clear than for the Big Five. Below, we present some of these results for each dimension. The barplots and wordcloud for the rest of the MBTI are available in the OSF.

- **Feeling - Thinking** (Figs 4 & 7): The Feeling dimension is characterized by words like 'feel', 'love', 'people', and 'feeling', aligning with MBTI's description of Feeling as prioritizing emotions. The Thinking dimension, however, while showing MBTI-consistent terms like 'logic', 'argument', and 'research', places a stronger emphasis on academic and scientific concepts.
- **Introversion - Extraversion**: Introversion emphasizes 'music', suggesting solitary activities, alongside introspective terms like 'mind' and 'think', however it also includes words like 'people' and 'world' with a less direct link to Introversion. Extraversion is led by 'people' and 'happy', aligning with traditional views of social relationships and positive affect. It features enthusiastic terms like 'awesome', and 'great', reflecting energy. Other terms are more ambiguous and less easy to assign to either dimension.
- **Judging - Perceiving**: Judging focuses on structure and planning, with top words like 'plan' and 'work', suggesting organized, goal-oriented approaches. Perceiving highlights spontaneity and flexibility, with 'yeah', 'fun', and 'music' dominating, indicating a more casual, open lifestyle. Notably, both of them share 'people' and 'time', suggesting either a difference in interpretation (commitments vs. fluid social engagements) or ambiguity in the classification.
- **Intuition - Sensing**: Intuition focuses on abstract concepts and big-picture thinking, with top word like 'life', 'world', and 'idea', suggesting a preference for theoretical and conceptual approaches. Sensing is more difficult to explain with its top words. Mostly they refer to concrete actions like 'ran', 'work', and 'help', focusing on tangible and immediate realities. Both sides of the dimension suggest a potentially ambiguous classification.

Given that the model is able to capture contextual information about the words, they should be analyzed in the original texts they belong to, so as to get a deeper understanding of how they contribute to the model's decision-making process. This will be presented in the next section.

## Explainability—whole text exploration

As detailed in the methods section, we employed three approaches to select the most significant texts: the highest overall text-level scores, highest attribution scores according to the geometric mean, and the highest according to the average. Among these, the geometric mean technique proved particularly effective in capturing true positives, providing a nuanced view of word importance in context. This method ensured texts that best exemplified the linguistic patterns associated with each personality trait.

Notably, the outputs from these three approaches largely overlapped, with only minor exceptions, suggesting a consistency in the model's focus across different approaches. This convergence lends additional confidence to the selected texts as representative of the model's decision-making patterns. The exemplar texts presented in Tables 6 and 7 are drawn from this

**Table 6. Example texts for Big Five traits.**

| Trait | Positive | Negative |
|---|---|---|
| Agreeableness | "I've been thinking about going on a mission for the church also…" | "Jessica Simpson is such a bitch, all through middle school me and my friends would have killed to get some * from her" |
| | "I forgive my mom for how she feels because she just doesn't know any better…" | "Will is annoying me. He has this dumb ass book he will not put down. I hate competing for his attention. Asshole. Oh yeah, Will is my boyfriend…" |
| | "I feel really bad about my dad having to pay so much money for me to get my education here. I want to help out more. I hate it when my parents have to put up money for me…" | "This guy just bumped into me and did not even apologize. I hate it when people do that. He just walks by like it's okay to knock people into a damn computer screen. Jerk…" |
| Conscientiousness | "I am pretty proud of myself today, for i got much more accomplished than I thought I would. I figured that after my last class I would just take a nap. But instead, I went to the gym and had a great workout…" | "I took French in high school and it was really tough. But maybe that was because I didn't really work hard my first two years of high school and I never really paid attention in class…" |
| | "It's always good to make a plan of everything that you want to do in a day. I always have to do lists. They are so useful, and I feel so organized when I make them. And when I finish everything on one list, I feel so good. It is a great feeling…" | "The tv is kind of annoying i want to turn it off but it's so far away. College is making me lazy…" |
| | "I have gotten all of my work for this week done already so I'm at ease about that…" | "I have to do one of those for English and I haven't even thought about. I'm a procrastinator it will probably be the night before and I hadn't even started on it…" |
| Extraversion | "I played basketball, ran track, softball, was a cheerleader all thourhg school, did student council, sisters of service, fellowship of christian athletes, and took a few leadership roles and it was great…" | "I like all my professors except my freshman seminar prof. He picks on me because I'm quiet and I don't talk much. I hate when teachers do that. It really bugs me. I like listening to people talk…" |
| | "My friends and I are all going to hardrock cafe that night to eat and then we might go to sixth street and party afterward…" | "I don't go out partying like most college students. I think I should take advantage of my college years here but I really don't like the whole clubbing experience. I'm full and I feel like vomiting…" |
| | "I ended up pledging alpha chi omega and I absolutely love it! The sweetest girls in the owrld are in there and they are so much fun!…" | "I am scared to meet people because they all seem so strange and their thoughts all seem to be the opposite of mine. I want to breathe a breath that is new […]. I am going to enjoy this class [psychology] even though is has about a billion people in it and that is a change since my town only has few people living in it…" |
| Neuroticism | "I sound like a goddamn theater major being all dramatic…" | "I love college […] I love this song […] I love the wide variety of people you get to see on a daily basis…" |
| | "Annoyance is about the only word I can think of at this moment to describe how I feel. Today has been awful. Why doesn't anything ever work out the way I wish it would? Everything I needed to accomplish today has been a filure…" | "After i do thins I may go and eat or spend some time with melissa, my girlfriend, who also happens to be my fiance she's great. After that I can write my router advertisement program. That would be fun. I love melissa. Being in austin is great. I like it much better that being in San Antonio…" |
| | "I have an eerie feeling that someone I know is loooking over my shoulder, watching me write this stuff that i wouldn't tell my closest friiends…" | "People told me that I would be overwhelmed when I came down here but I haven't felt that way yet. Things are getting harder by the week, but I think I am handling it well…" |
| Openness to Experience | "I had my audition for the "madrigal dinner" tonight. I think it went pretty well. I sang a song that I wrote…" | "I am quite addicted to diet cokes and diet drinks in all, I think I have a serious problem, yet I live for it. I feel like one right now. Boy do I need to do laundry. It just keeps piling up, of course the one day I finally decide to do it, everyone else has the same idea. I can't wait to go through the dorm experience, and then move on into an apartment and have things of my own…" |
| | "I then imagined as though I were past the class, but somehow not in the real world as we know it, but somewhere where the sword still ruled the land. I think now it must have been my own idea of feudal Japan, but needlessly I walked around in nothing but a (I forget now what the karate uniforms I have donned so many times are called) but I was wearing one of those, carrying a real sword this time, trudging through muddy roads through a country side constantly lit by an orange, pasty sun. I can't remember where it went form there, but needless to say much violence, honor, and success followed. That's how my daydreams sometimes run, but sometime they are more erratic…" | "I haven't been to a beach in two summers now because of the knee surgery and rehab (for the knee), and this summer we were just too busy. That reminds me, I miss my family in Virginia. Maybe I can fly out there this summer. That would be fun. Virginia is beautiful. The Chesapeake area especially. I'm glad my brother is alright, but when my mother called it worried me. But he will be fine. I cleaned the guest bathroom tonight (I have to do chores around the house because I live in a co-op)…" |
| | "Ellipses are fun, they provide space, and soetimes a depth that no other literary device can reach. I'm not sure if depth is the right word of if tis truly a literary device, but ellipses are (thoughts search for word) some-thing that people use. Woah, a mental stumble in words, I wonder how oftne I do that?…" | "I left my medicine at home so I'm having trouble breathing. But my mom is sending it to me so I can feel better soon. I have to go to a UT football game for an assignment in my freshman seminar class but I didn't buy a sports package which by the way I think is the stupidest think I've ever heard of. I think if you're a student then you should be able to get into the game for free but what do I know…" |

**Table 7. Example texts for MBTI dimensions.**

| Type | Label 1 | Label 0 |
|---|---|---|
| I/E | [...] This is how the poem relates to me: I am so often swept up in my internal world. Always seeking my passions. I love them more than anything. But sometimes I turn around and realize that there is...||| I get strong scores for 3, 4 and 5 and can relate to all of them. Especially 4w3, 3w4, 4w5 and 5w4. When I first did an enneagram I was amazed on how accurate those four descriptions were (and I...||| Score: 30 10char||| I can relate to everything in your post. I'm also a 24-year old INFP Christian, and I struggle to find out how to reach my dreams despite my introversion. I'm 100 % sure that we can change our...||| First. This post is NOT about supernatural abilities or spiritual gifts. (we could open another thread for that). I remember that someone on this forum wrote that we INFP have the uncanny...||| | How can you possibly choose?!?!....... I suppose The Rolling Stones, but this is so hard!||| Hell no! I'm the creepy closet lurker :[ *sets up a surveillance camera in your room for when you get back from my...||| You had me at What the fuck are you doing in my closet? <3||| Congrats!! :kitteh:||| Bans you because the customer is always right!||| I've ben here about 8 months now, so I selected 6 months.||| Definitely the mafia sub-forum if that were an option, but overall my favorite section is the entertainment plaza. The Personality Test Resource section is a close second.||| Yay! Congrats MindSlinger !!! :kitteh:||| I lost my teddy bear D:.... Can I cuddle with you instead?||| |
| N/S | I didnt see where I said anything about jealous- and no Christians dont refer to his as evil- um yeah- what are you 5 years old....||| entjwillruletheworld Since the post was the last was one I didnt feel a need to post as I have in the others- hannibal- said how did intro turn into a war end quote||| I understand what you are going threw. After my divorce I questioned myself alot and still do at times, with many of the questions you asked on here. I felt I had put so much into and made many...||| [...] child would die and how he would...||| And yes it was made into many religions- but it was taught in the bible that there would be many false religions and the bibe typically speaks of christians | always talk to strangers, no matter how creepy they are. attempt to pick pocket as many people as possible. telling your boss off is a good way to blow off some steam. ||| think he was terrible strategically. he could have destroyed the bef at dunkirk but decided to let them evacuate. then hitler decides to shift the battle of britain to bombing london hoping they... ||| spaghetti and garlic bread. ||| welcome to the forum. ||| welcome to the forum. ||| welcome to perc. it took me a couple tries to get a better understanding of which type i am. ||| exactly. a good looking person for example could still be insecure. ||| physically or mentally ? ||| isfj and i add two spoons of sugar and two of cream. ||| [...] ||| welcome to the forum. ||| welcome to the forum. ||| welcome to the forum. ||| understand that worry is just made up thoughts in your head that you created. ||| my dad's good one liner you always need to take care |
| F/T | [...] Faeriegal has a great post (and personal experience!)..but I thought I'd share my thoughts as well. I used to consider nursing...and I'm in the same boat as you are with trying to decide my life...||| I am curious why you think Drizzt would be INFJ. He has a very strong Fi (rather than Fe) function in my opinion. I've always thought him to be INFP/ISFP. What makes him INFJ? :)||| [>.<Sorry, dunno how to delete this.]||| I am obsessed with the concept of being as in tune with your natural and true self as possible. The ideals of society aggravate me; how people can be so blind...and shape themselves to society's...||| I read that Captain Ahab in Moby Dick is INFP. :) | [...] Yes, I like high standards but it's just not practical enough. If I'm attracted to a woman then I just am. I didn't selectively take the time to think about it and choose to be. My brain fired off my...||| Yes, probably someone purposely told others that it's a brain test and people believed it. Probably a bad ENTJ told someone that it was a brain test, okay which one of you ENTJ's is responsible for...||| People actually misinterpret the spinning lady as a left brain/right brain form of test. It is actually created as an optical illusion and often mistaken as a test. [...] |
| J/P | ESTJ's get all the credit, but ESFJ's do the same thing WHILE managing the emotional...||| Interestingly, we did have some fights where I finally blew my lid, and I acted in a way that I was HORRIFIED by after the fact (honestly, even as I was doing it, but I felt completely out of control...||| I can second this experience (INTJ male living with ESFJ female). We had some conflict for a while after we moved in together simply because her tone was received (by me) as being much more...||| The key to any type re-finding their center and regaining balance is the auxiliary function—in this case for an INTJ, that Extraverted Thinking. He needs to (at some point) stop dwelling on how he... [...] | [...] Cmon, who butters their toasts, honestly... Also, I don't mean to question your judgement or anything, but are you sure he's not ENFJ? I can't really see ENFP's being fluffy since our F is...||| I knew she was either INFJ or ISFJ (introverted Fe users). My best friend was an INFJ too, I've found out that the ENFP is the doormat in these relationships. I don't know why, but when you frickle...||| I'm finding Johnny Depp the most obvious famous Ne user. But I'm pretty sure he's more ENTP than ENFP.||| @MuChApArAdOx Yes, I also view ENFP women as more alpha than most other types, but only because you're generally the more feminine women (which goes hand in hand with the (E)NF temperament), not... |
| Masked dataset | | |
| I/E | been in any serious situation. ||| alone, away from everyone. can't see anyone, can't hear anyone, can't sense anyone. ||| I wake up early and think for a few hours. If I'm worried about something I'll wake up super early, check the thing I'm worried about, and go back to bed to think. Then I get out of bed, sit on the... ||| I don't dislike them, not really. I don't really talk to them, but I think that's more out of respect for their personal space. ||| Something that could be a part of it is the fact that they are one of the most common types, the most common in females. In today's snow flake obsessed society, it seems that if there are lots of... | you do seem like an [UNK] judging by your posts ||| b: [UNK] m: [UNK] k: [UNK] [UNK], [UNK], [UNK] ||| [UNK] type w8? interesting i am not sure maybe you're an [UNK] ||| k : [UNK] m : [UNK] b : [UNK] [UNK], [UNK], [UNK] ||| k m : [UNK] b : [UNK] [UNK], [UNK], [UNK] ||| it's just a vast difference of interests intelligent people are usually interested in things difficult to understand not because of how difficult they are but because the things difficult to... ||| the guy seems like an [UNK] at first glance so manipulative and a great con who is con at getting people do what he wants to do however i can't help but think [UNK] since he seems to have that [UNK] vibe...[...] |

*(Continued)*

**Table 7.** (Continued)

| Type | Label 1 | Label 0 |
|------|---------|---------|
| N/S | Well, Germany is absolutely and unquestionably a social culture, though the prussian character is fixed. Switzerland is the German-speaking country, Austria is. About Germany there is... \|\|\| yes, I found out from a mutual acquaintance that this is true—bliss stream owns stack me up. It brings up a significant issue, in that I had found his sp/sxw1 description to describe someone else I... \|\|\| I decided to actually read what Beatrice chestnut has to say about the place of shame in the image (sorry, sadness) triad. I'm finding chestnut's book to be a lot more valuable when she just focuses... [...] | [...] i'm sure this goes beyond personality and there's most likely more to her story as to why she bahaves (ed) that way. anyone can have an angry disposition regardless of their personality. \|\|\| zombie, you're one of my faves on perc. see, you're an [UNK] and i don't dislike you. for the most part, [UNK] are peacemakers so we will default to the most practical and efficient way of... \|\|\| i'm really bummed my post looks like that. and when i tried to edit it, it wouldn't save it. i gave up. sorry hunny. [...] |
| F/T | are many advisers within me: ratio/logic, psyche and emotional needs, morals, expectations of other/close people, bodily needs/demands, worldview and mindsets, gut feelings, intuition,... \|\|\| I believe [UNK] with well developed [UNK]...—have left behind the victim identity—are not only sincere and authentic but also competent—Don't only desire to make a change in the world, but... \|\|\| I hate when I have a fight with my imaginary wife and then feel down all day... \|\|\| I'm horrified at how quickly I can lose the connection to my heart, to the real core of who I am and what truly matters. There are so many ways to spend my time and C can get so caught up in the... | if you do the person will probably just blow up in your face.... \|\|\| so if you consider a man an animal and humans are animals, what does that make you ? you're not an alien are you ? : crazy : yes, you could say the same thing with sex. there are consequences to it... \|\|\| yes, i am subconsciously attracted to good hips and thighs. the measurement is somewhere along the lines of this girl, she has fantastic hips and thighs. the hips and thighs theory is not bs, i am... \|\|\| yes, i like high standards but it's just not practical enough. if i'm attracted to a woman then i just am. i didn't selectively take the time to think about it and choose to be. my brain fired off my... [...] |
| J/P | [...] [UNK] are fun to be around. I greatly enjoy their company. In fact, I really really like an [UNK] but he mentioned once that the feeling was not mutual. I left it as is. Not to say, that I gave up or... \|\|\| Why, of course! you know he love you. Since we love the person, we don't mind. We just laugh at ourselves inside as well. \|\|\| I totally agree with the snapping. I have been called out on this numerous times at work especially. It is just part of me. The death stare as well. \|\|\| [UNK] usually don't keep close connection with their ex's. I don't. It took me almost five years to change my mindset about my [UNK] ex. He doesn't use harsh words but he takes action in cold and... [...] | crook who is about as smart as a box of rocks and police who aren't much smarter.... a few... \|\|\| i used to feel like a zombie, now i just find ways to scare the shit out of myself regularly. usually snowboarding or rock climbing. a lot of times just getting out and fishing or even offroading... \|\|\| my left ear is completely clogged. not being able to hear out of one ear is the most annoying thing ever. \|\|\| i once had one night stands in a row. the first two were great. i just got up in the morning, said bye, and walked home. the third was pretty uncomfortable because i woke up alone on the couch... [...] |

comprehensive selection process, offering a robust representation of how our model interprets and classifies personality traits and types based on textual data. A full version of these examples are in Supplemental Materials 2 (S2 File) and in the OSF organized in folders.

For the Big Five traits, we identified texts with high attribution scores that exemplified or contradicted each trait. Table 6 presents a condensed selection of these texts, showcasing how the model distinguishes between language patterns associated with high and low levels of each trait. These texts illustrate the linguistic patterns and content that the model found most indicative of each trait or type, providing concrete examples of the features driving the classification decisions.

Analysis of these high-scoring texts revealed clear linguistic markers for Agreeableness, Extraversion, Neuroticism, and Openness to Experience. Conscientiousness, however, proved more challenging to identify, with clear examples appearing only in isolated cases. These findings support H3, demonstrating that the words and their context most influential in Big Five classification are often semantically coherent with the trait's content.

While many high-scoring words aligned directly with the semantic content of the traits, our analysis also revealed more nuanced patterns that initially seemed counterintuitive. For example, Agreeableness showed words like 'hate' with high attribution scores which seemingly does not have coherence with its meaning. Closer examination revealed these instances often occurred in contexts that reinforced agreeable tendencies, like 'I hate it when my parents

have to put up money for me' or 'I hate to admit it but I really do miss them'. These findings, that were found across traits, underscore the importance of considering words within their broader context and demonstrate the model's ability to capture subtle linguistic nuances beyond word choice. This contextual sensitivity further supports H3, showing that the model identifies semantically similar language used in trait-coherent versus trait-incoherent ways.

The MBTI dataset analysis (Table 7) highlighted stark differences between the original and masked version. In the original dataset, high-scoring texts often contained explicit mentions of MBTI types, supporting our earlier findings about self-reference in prediction. The masked dataset results (full version in the OSF) showed how the model adapted to more subtle linguistic cues when explicitly type indicators were removed. For F/T, for instance, many texts contained words related to feelings and thinking styles. Importantly, in the masked results, the texts were showing the masking term we are using ('UNK'), signaling the strong influence of the types terminology on the classification. The less clear results for the MBTI when compared to the Big Five, demonstrate stronger coherence with theory in the Essays dataset and the Big Five Theory.

These contextual examples provide crucial insights into how our models interpret personality traits and types in text. They demonstrate that the model captures nuanced expressions of personality that go beyond simple word usage, often identifying complex patterns of language use associated with different personality traits, but being less clear for types. This visualization analysis complements our word-level findings by illustrating how individual words contribute to broader linguistic patterns indicative of personality traits.

## Discussion

In this study, we introduce a novel approach on text-based personality research that works with unstructured text, leveraging AI explainability techniques. Moving beyond mere performance metrics (accuracy or AUC) improvements, we identify influential parts of the input text to examine the inner workings of the algorithms. This approach allows us to explore the data considered for text classification, improving the content validity of the LLM-based approach [24]. By analyzing unstructured language data, through this lens of explainability, we aim to ensure the replicability of two personality theories in text [24,94] and understand the algorithm's decision-making process in a more naturalistic context.

We found accuracies and AUCs for the two datasets in the classification task, close to what has been reported in prior studies [18,34,52,55,57,69,70,83,92] (see Tables 1 and 2 in S1 File). Notably, proper data partitioning and fine-tuning serves as warranty for good classification performance.

An exhaustive review of the texts identified as significant by the explainability algorithm revealed abundant evidence of theory coherence between the Essays texts and questionnaire items for most traits, with Conscientiousness being the exception. For the MBTI dataset, our findings confirmed H4, showing that in the original data, posts contained explicit typologies, while masked data results were more ambiguous.

Our analysis also revealed the model's nuanced understanding of language in personality assessment. Words seemingly incongruent with a trait often received high attribution scores in trait-coherent contexts (e.g. the word 'hate' for Agreeableness). This contextual sensitivity, observed across traits, demonstrates the model's ability to distinguish between trait-coherent and trait-incoherent language use, further supporting the theory coherence found in the Essays dataset. This nuanced interpretation was less evident in the MBTI dataset. These findings highlight the need to revisit past studies that relied on techniques like the LIWC [2,

3,64], as contextual embeddings from our models may uncover more nuanced patterns of personality and psychological expression in text.

The stark contrast between the effects of targeted and random masking on our MBTI classification model provides compelling evidence for the self-referential nature of personality type discussions in the Personality Café forum, supporting H4. The model's reliance on explicit mentions of personality types, rather than subtle linguistic cues, would be in line with what Stein and Swan [79] pointed out, where conventional tests made to determine the MBTI type of individuals would indicate their knowledge and preference of MBTI types rather than their underlying personality characteristics. This aligns with MBTI theory's internal consistency problems and indicates a potential self-fulfilling prophecy where people reinforce their typological identities through language. However, this phenomenon may be amplified by the dataset's selection bias, as participants willingly engage in type-focused discussions. While some texts, particularly in the F/T category, showed coherence with dimensions theory, these were exceptions rather than the norm.

While studies comparing the Big Five and the MBTI using NLP approaches are limited, our findings align with previous research. Notably, Celli and Lepri [26] reported similar results in classification tasks and a preference of the Big Five over the MBTI, albeit using less advanced techniques. Our study extends these findings, employing more sophisticated NLP methods to further validate these earlier observations.

These results underscore the importance of critically evaluating the construct validity of personality assessments derived from text data [85], especially when dealing with popular but scientifically contested models like the MBTI. Furthermore, they highlight the potential of NLP techniques not just for personality assessment, but also for uncovering the ways in which personality theories shape discourse and self-presentation.

Our study also highlights methodological concerns in personality assessment via NLP. The dichotomous classification of the Essays dataset, following Oberlander and Nowson [60], is problematic: using the mean as a threshold to delineate the two groups [51]. This classification methodology leads to minimal differences between individuals with varying scores [60], contradicting the continuous nature of personality traits and favoring an artificial dichotomy. This artificial dichotomy might have contributed to lower accuracy scores and affected the explainability of results, particularly for traits like Conscientiousness, suggesting the need for more research into this trait [72] Efforts to perform these analyses with the correct type of data should be pursued. Studies employing alternative datasets have been more attentive to this key consideration [71,82].

The use of the MBTI dataset, despite its questionable psychometric validity [37,67] and extensive criticism [23,54,79], is justified by its widespread use and societal impact [55,56]. We demonstrate that the seemingly better performance of MBTI over Big Five classification is largely due to reporting accuracies instead of AUC and not accounting for the use of MBTI-related terminology. This conclusion alone is key to encourage the field for research beyond performance metrics in classification tasks for personality psychology [24,27].

Our study contributes significantly to making personality detection more accountable and transparent. By going beyond performance metrics and exploring the decision-making processes of our models through explainability, we have uncovered important insights about the nature of personality expression in text and the limitations of current assessment methods. This analysis not only highlights the need for a contextualized approach to explainability results in text data, but also underscores the potential of NLP techniques to uncover subtle linguistic markers of personality that might not be immediately apparent from traditional trait descriptions and less advanced computerized techniques [5]. These findings not only

advance our understanding of personality assessment through NLP but also highlight the complex interplay between personality theories, individual self-perception, and language use.

## Limitations

One important question in the integration of personality psychology and NLP is the quality of the prompts used to elicit data in humans or generative agents alike. For our datasets, both generated by humans, we found biases implicit to each scenario.

The Essays dataset should be used considering that it might not be the best prompt to detect personality traits as low-level resolution constructs, or capturing the multi-faceted components expressed in their facets and nuances at a high-level resolution. It might be better suited to capture the signal of other psychological constructs (e.g. impulsivity, disinhibition, sensation seeking, etc.). Conversely, the MBTI dataset, sourced from an online community deeply interested in personality theories, likely influences selection bias. Of note, although online data might appear more authentic, our results suggest that stream-of-consciousness prompts can effectively elicit authentic data, in line with previous related research [53].

Another important limitation in our two datasets, is the inability to use external criteria for validation, as is the golden standard in psychometry [24]. In this sense, explainability results should be associated with behaviors and life outcomes as established in the field of personality psychology [61,80].

The use of dichotomous classification for the Essays dataset is problematic. This approach contradicts the continuous nature of personality traits and may have contributed to lower accuracy scores and affected the explainability of results, particularly for traits like Conscientiousness.

Our interpretation of explainability had one limitation: while comprehensive, our approach to evaluating coherence between explainability outputs and personality theories was subjective, as it involved human evaluation, in contrast to the rest of the methodology employed, which was driven by data.

Finally, another limitation of our work is related to the explainability technique. Commonly used techniques, including the one used here, can only produce information for individual features —words in our case— of the input. This makes the interpretation of results more difficult, since the semantic aspects of a particular word are strongly influenced by the surrounding words. Explainability techniques such as the one proposed by Sikdar et al. [77], are able to capture information about the interaction between features or words, constituting a very interesting option to extend the work presented here.

## Future directions

Analysis of personality-labeled texts with LLM should be extended to other types of texts. This study applied it to stream-of-consciousness texts and web posts that although useful might not be the best option to elicit words and topics with personality signals. The data collected from Facebook has shown better scores in all the metrics scores for all traits [28] suggesting that the fidelity of the output is directly correlated to the quality of the input. Generally, it appears that data collected from social media contain higher levels of self-disclosure than other datasets [59], although the evidence for this is mixed [53].

However, it would be insightful to study the same dataset without any pre-established structure, adopting a more exploratory approach. This method aims to uncover other psychological processes that may influence the production of spontaneous texts. This exploration could also yield valuable insights into the underlying mechanisms of personality expression

in unstructured writing and might provide information for a more complex theory of personality.

Further confirmation of H4 in the MBTI datasets should be conducted with a dataset that does not rely completely on topics on personality theories. The PANDORA Talks dataset [39] could be a valuable resource for this purpose.

Many authors have used the labeled Essays dataset without proper consideration of its origins and with misconceptions about its construction. Ultimately, we are predicting labels that are questionably assigned to the essays. Future studies should move beyond dichotomous labels for traits that are not founded in sound research [60] and explore personality prediction with the tools presented in this study, thus respecting the continuous nature of personality traits [82].

One exciting avenue for research is to push the lexical hypothesis out of its current boundaries. While adjectives have been extensively studied to build a taxonomy of personality, more complex language productions like whole sentences or even paragraphs would also hold valuable personality information —e.g. The Characters of Theophrastus [88]. For instance, a sentence like "He always arrives early to meetings and has a detailed agenda prepared" might capture conscientiousness more comprehensively than single adjectives like "organized" or "punctual". The Essays dataset has not been explored in this direction, and using sentences and paragraphs with their respective embeddings could help build a more comprehensive personality taxonomy, as suggested by early personality researchers [21].

Future research should also examine differences in trait expressions across demographic variables. For instance, women and men might have different ways of expressing through verbal behavior their extraversion or introversion [65]. Additionally, predicting life outcomes, such as academic performance or subjective well-being should be considered [82].

Future research should focus on developing computerized evaluation approaches for explainability techniques in personality assessment, with objective NLP methods. Further exploration of explainability data using quantitative approaches can reduce human interpretation and enhance objectivity when analyzing results within the framework of personality theory (e.g. given a set of words identified by the explainability technique, use a data-driven technique to infer the trait that better explains the set of words). These approaches would significantly advance the field by establishing consistent, data-driven methods for analyzing explainability outputs with minimal human intervention.

We believe that the introduction of the toolbox of techniques from the field of NLP aligns with the fundamental tenets of the lexical hypothesis, as it facilitates a contextualized analysis of text beyond individual words. This approach also enables the examination of longer text strings, offering a more comprehensive understanding of language usage. While further research in this direction is warranted, it is essential to validate LLMs within the current personality psychology framework before addressing these challenges. Our study was designed with this notion in mind.

Some authors suggest that LLMs will eventually replace questionnaires [34]. However, we are still far from dispensing with questionnaires entirely and their influence is still present even in NLP/PA studies despite concerns on using them as ground truth for personality research [8]. Future efforts in our field should first expand the current theory to move beyond questionnaires and these efforts should integrate not only textual data but multimodal and behavioral data [29]. This could involve incorporating behavioral observations, peer reports, digital behaviors and other objective measures alongside language data.

Additionally, NLP/PA and TPA should at some point be compared in terms of the variance they explain in predicting personality, to reveal the relative strengths and limitations of each

approach. This could potentially lead to more integrative and robust models of personality assessment.

Ultimately, our study contributes to the effort of making personality automatically detected with better accountability of the processes behind each classification decision. Although it raises ethical concerns, automatic personality can be used for improving personal assistants and generative agents, allowing the creation of more intelligent empathetic systems, personalizing experiences and suggestions, job screening, political forecasting, forensics [56], and ultimately influence the pursuit and measurement of social change outcomes [38].

## Acknowledgements

The first author thanks the members of the IDLab and the HSG-IBT research teams for their valuable feedback, time, and discussions.

## Supporting information

**S1 File. Contains Supporting Tables 1–4, Supporting Figures 1–16 and Annex with a list of MBTI types and related terms used in masking and analysis.**
(DOCX)

**S2 File. Contains Annex with indicative text from the Big Five traits and MBTI types.**
(DOCX)

## Author contributions

**Conceptualization:** David Saeteros, David Gallardo-Pujol, Daniel Ortiz-Martínez.

**Data curation:** David Saeteros, Daniel Ortiz-Martínez.

**Formal analysis:** David Saeteros, Daniel Ortiz-Martínez.

**Funding acquisition:** David Gallardo-Pujol.

**Investigation:** David Saeteros, David Gallardo-Pujol, Daniel Ortiz-Martínez.

**Methodology:** David Saeteros, Daniel Ortiz-Martínez.

**Supervision:** David Gallardo-Pujol, Daniel Ortiz-Martínez.

**Visualization:** David Saeteros, Daniel Ortiz-Martínez.

**Writing – original draft:** David Saeteros, David Gallardo-Pujol, Daniel Ortiz-Martínez.

**Writing – review & editing:** David Saeteros, David Gallardo-Pujol, Daniel Ortiz-Martínez.

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
