## [Decision Letter · Decision Letter 0]

PONE-D-24-37721 Text Speaks Louder: Insights into Personality from Natural Language Processing PLOS ONE

Dear Dr. Ortiz-Martínez,

Thank you for submitting your manuscript to PLOS ONE. After careful consideration, we feel that it has merit but does not fully meet PLOS ONE’s publication criteria as it currently stands. Therefore, we invite you to submit a revised version of the manuscript that addresses the points raised during the review process.

We look forward to receiving your revised manuscript.

Kind regards,

Vijaya Prakash Rajanala

Academic Editor

PLOS ONE

“This research was supported by the 2021SGR0709 grant from the Government of Catalonia and Grant PID2020-119755GB-I00 funded by MCIN/AEI/10.13039/501100011033. The authors declare no conflict of interest.”

“This research was supported by the Government of Catalonia (https://govern.cat/gov/), Grant number 2021SGR0709 received by DGP, and by the Ministry of Science and Innovation of Spain, MCIN/AEI/ 10.13039/501100011033 (https://www.ciencia.gob.es/), Grant number PID2020-119755GB-I00 received by DGP. The funder had no role in the study design, data collection and analysis, decision to publish, or preparation of the manuscript.”

3. We note that Figures 2, 4 and 6 in your submission contain copyrighted images. All PLOS content is published under the Creative Commons Attribution License (CC BY 4.0), which means that the manuscript, images, and Supporting Information files will be freely available online, and any third party is permitted to access, download, copy, distribute, and use these materials in any way, even commercially, with proper attribution. For more information, see our copyright guidelines: http://journals.plos.org/plosone/s/licenses-and-copyright.

1. You may seek permission from the original copyright holder of Figures 2, 4 and 6 to publish the content specifically under the CC BY 4.0 license.

Additional Editor Comments (if provided):

The second reviewer suggested referencing her own paper; the author may disregard these suggestions.

The author had to meticulously examine the reviews and revise the paper.

Reviewers' comments:

Reviewer's Responses to Questions

**Comments to the Author**

1. Is the manuscript technically sound, and do the data support the conclusions?

Reviewer #1: Yes

Reviewer #2: Yes

2. Has the statistical analysis been performed appropriately and rigorously? 

Reviewer #1: I Don't Know

Reviewer #2: No

3. Have the authors made all data underlying the findings in their manuscript fully available?

Reviewer #1: Yes

Reviewer #2: Yes

4. Is the manuscript presented in an intelligible fashion and written in standard English?

Reviewer #1: No

Reviewer #2: Yes

5. Review Comments to the Author

Reviewer #1: I don't understand the authors' motivations. Why test the effectiveness of these four types of feature extraction methods in the field of personality prediction, without studying methods with higher predictive performance

The paper needs to be reconstructed, and the current version is difficult to read.

Compared to the References62, this paper only explores three additional methods: IDP, Integrated Gradients, and Word Clouds. I think the lack of innovation in this paper supports its publication on Plos One.

Reviewer #2: The paper presents an interesting and important topic. However, there are several areas that could be improved:

Introduction: The introduction is too lengthy and may lose the reader's interest. Consider condensing the content while maintaining clarity.

Typos & Formatting:

- There are multiple instances of incorrect apostrophe usage, such as in ‘INFP’. Please correct all occurrences.

- Similarly, in phrases like ”This guy just bumped”, the apostrophe usage should be revised.

- The selection of models and evaluation metrics is not particularly novel. Did the authors consider using an LLM-based baseline for comparison?

- If there are novel findings, they should be highlighted and discussed explicitly.

- The limitations section should include a discussion on this aspect.

- Word clouds are generally not recommended for scientific papers. Consider replacing them with more rigorous visualizations, such as bar charts or heatmaps.

The literature review should be updated to cover bias-related issues in these models. Which aspect of NLP is this paper focusing more? it should be mentioned, is it ner/ nli/ token classification, the literature while focusing on big 5 traits theory, should shed some light on the methods selections (though brief), I am also curious about choice of metrics by authors.

- Additionally, the study appears to be confined to a single model family. Since different models generate probabilities differently, discussing these variations and their implications would strengthen the paper.

6. PLOS authors have the option to publish the peer review history of their article (what does this mean?). If published, this will include your full peer review and any attached files.

Reviewer #1: No

Reviewer #2: No

---

## [Author Response · Author response to Decision Letter 1]

11 Mar 2025

Department of Mathematics and Computer Science

Universitat de Barcelona

Barcelona, Spain

Thursday 27th February, 2025

Academic Editor

PLOS ONE

Dear Dr. Rajanala,

Thank you for your feedback on our manuscript "Text Speaks Louder: Insights into Personality from Natural Language Processing" (PONE-D-24-37721). We appreciate the constructive comments from you and the reviewers.

Regarding the journal requirements:

1. Regarding the figures' copyright: Figures 2, 4, and 6 are original work created by the authors and are already under CC BY 4.0 license. No additional permissions are needed as we are the copyright holders. All figures and supplementary materials are available in our OSF repository, which is referenced in the manuscript and shared under the same CC BY 4.0 license.

2. As requested, I have removed funding information from the Acknowledgments section. The Funding Statement remains unchanged and should read as follows: "This research was supported by the Government of Catalonia (https://govern.cat/gov/), Grant number 2021SGR0709 received by DGP, and by the Ministry of Science and Innovation of Spain, MCIN/AEI/10.13039/501100011033 (https://www.ciencia.gob.es/), Grant number PID2020-119755GBI00 received by DGP. The funder had no role in the study design, data collection and analysis, decision to publish, or preparation of the manuscript."

[PLACEHOLDER: PLOS ONE style requirements]

Regarding the reviewers' comments:

Reviewer #1: "I don't understand the authors' motivations. Why test the effectiveness of these four types of feature extraction methods in the field of personality prediction, without studying methods with higher predictive performance? The paper needs to be reconstructed, and the current version is difficult to read. Compared to the References62, this paper only explores three additional methods: IDP, Integrated Gradients, and Word Clouds. I think the lack of innovation in this paper supports its publication on Plos One."

We appreciate Reviewer 1's comments and would like to clarify our research objectives. Our study's novelty lies in how we use these models to study the relationship between language use and psychological traits - a fundamentally different approach from previous work that focused primarily on classification performance. This enables us to understand not just whether these models can predict personality traits, but how they make these predictions and whether their decision-making aligns with established psychological theory. The novelty of our work cannot be evaluated solely through accuracy metrics, as our contribution lies in explaining and validating model decisions through the lens of personality theory. This focus on explainability over performance is intentional and important.

While numerous studies have focused on improving accuracy metrics (as documented in Tables 1 & 2 in our Supplementary Materials) including Reference62, very few have examined whether these improvements stem from theoretically meaningful patterns in language use or from artifacts in the data. High accuracy scores alone do not guarantee that a model is capturing genuine personality signals - they might instead reflect dataset artifacts or spurious correlations. Our study specifically addresses this critical gap by providing a systematic framework for evaluating whether model predictions align with established personality theory, regardless of their raw performance metrics.

Regarding the paper's readability, we have restructured the introduction to better clarify our objectives and approach. In particular, we have revised the section discussing the lexical hypothesis and the relationship between traditional personality assessment and NLP-based approaches, as well as a key paragraph right before our research hypotheses that now better emphasizes how we leverage NLP models to examine personality manifestation in natural language, rather than focusing on prediction performance. These revisions make our contribution more explicit and our methodological choices more clearly motivated.

Reviewer #2: We appreciate Reviewer 2's thoughtful feedback and have addressed each point:

"The introduction is too lengthy and may lose the reader's interest. Consider condensing the content while maintaining clarity."

We acknowledge the reviewer's concern about the introduction's length. Following this feedback, we have substantially condensed this section, reducing it by approximately one full page while maintaining the essential theoretical foundation and literature review. Key revisions focus on tightening the argument structure.

"There are multiple instances of incorrect apostrophe usage, such as in 'INFP'. Please correct all occurrences. Similarly, in phrases like 'This guy just bumped', the apostrophe usage should be revised."

The apparent typographical errors and non-standard apostrophe usage appear primarily in direct quotations from our datasets (Essays and MBTI forum posts). We deliberately preserved these texts in their original form as this is exactly what our language models processed. Modifying these would misrepresent the actual input analyzed by our models. We have, however, thoroughly reviewed and corrected any apostrophe usage in our own scholarly writing.

"The selection of models and evaluation metrics is not particularly novel. Did the authors consider using an LLM-based baseline for comparison?"

Our choice of BERT and RoBERTa was deliberate and aligns with our research objectives. While newer models like DeBERTa exist and might achieve marginally higher performance, their substantially higher computational requirements would have necessitated aggressive text truncation, potentially compromising our analysis of long-form texts. More importantly, our paper's novelty lies not in the selection of models but in how we use them to study the relationship between language use and psychological traits - a fundamentally different approach from previous work that focused primarily on classification tasks.

Our implementation differs from prior research in several key ways:

• Instead of treating these models as black-box classifiers, we leverage them to understand how personality traits manifest in language use

• We focus on explaining model decisions and validating them against personality theory rather than optimizing for accuracy

• We demonstrate how explainability techniques can reveal meaningful patterns in language that align with psychological constructs

• Our approach prioritizes theoretical validation over raw performance metrics

These widely-used, open-source models were chosen because they:

• Allow for transparent examination of decision-making processes

• Achieve comparable performance to previous studies

• Support our chosen explainability techniques

• Ensure reproducibility of our findings

We first validated that our implementation achieved similar (see Table 1 of SM1 here) accuracies to previous work before proceeding with our novel explainability analysis. This foundation allowed us to focus on our main contribution: understanding how language models capture and process personality-relevant information in text.

"If there are novel findings, they should be highlighted and discussed explicitly."

Our key contributions are presented in the Results section, particularly in our explainability analysis. These include:

• Identification of specific patterns in language use that correspond to different personality traits, revealing how individuals naturally express personality through text.

• Demonstration of theory coherence between Big Five traits and model attention patterns.

• Revelation of self-reference effects in MBTI classification.

• Novel insights into how language models process personality-relevant information.

"Word clouds are generally not recommended for scientific papers. Consider replacing them with more rigorous visualizations, such as bar charts or heatmaps."

We fully agree with the reviewer's concern about relying solely on word clouds in scientific papers. In our manuscript, we had already addressed this by consistently pairing each word cloud with its corresponding bar plot, as both visualizations display the same geometric mean values from our attribution analysis. While bar plots excel at precise numerical comparisons, word clouds enable rapid pattern recognition of word relationships. To further emphasize traditional scientific visualization methods, as suggested by reviewer 2, we have:

• Reorganized our presentation of results to prioritize bar plots as the primary visualization method

• Renumbered our figures to lead with bar plots (now Figures 2, 4, and 6) followed by their corresponding word clouds (now Figures 3, 5, and 7)

• Rewritten the relevant sections to emphasize the quantitative analysis provided by the bar plots while maintaining word clouds as complementary visualizations

• Maintained both visualization types as they serve different but complementary purposes.

For this changes, check Explainability - Word Attribution Scores section.

"The literature review should be updated to cover bias-related issues in these models. Which aspect of NLP is this paper focusing more? it should be mentioned, is it ner/ nli/ token classification, the literature while focusing on big 5 traits theory, should shed some light on the methods selections (though brief), I am also curious about choice of metrics by authors."

We appreciate the reviewer's suggestions. Our paper focuses on analyzing how language use relates to individual personality traits. To achieve this goal, we apply explainability techniques to a classifier that predicts the presence or absence of each trait. While classification tasks play an important role in our study, they serve as a tool for our broader research objectives rather than being an end in themselves. The methods section thoroughly explains our methodological choices:

• BERT and RoBERTa for classification, due to their demonstrated performance in text classification tasks and public availability ensuring reproducibility

• Integrated gradients for explainability, chosen for its axiomatically justified properties and efficient calculation

• Multiple visualization techniques to present attribution results

While NER and NLI are important NLP tasks, they are not relevant to our specific research question of predicting personality traits from text and understanding how these predictions are made. Our paper addresses biases extensively, particularly in the MBTI dataset analysis where we demonstrate how self-reference affects classification performance.

"Additionally, the study appears to be confined to a single model family. Since different models generate probabilities differently, discussing these variations and their implications would strengthen the paper"

While examining multiple model families could provide additional insights, our focus on transformer-based models was driven by the fact that transformer models currently constitute the state of the art in language modeling. Within the transformer architecture, we experimented with two different model families: BERT and RoBERTa. The methodology section (Materials and methods: Analytic Techniques) explains the differences between these families, and the results section (Results: Accuracies and AUCs) shows the performance obtained by both models during classification. Despite RoBERTa being a more refined model compared to BERT, our experiments did not show significant differences between them. We chose not to conduct experiments with newer families like DeBERTa due to the prohibitively high computational resources required to process texts composed of hundreds of words. The selected models, in combination with explainability techniques, provide a robust foundation for understanding how these models make decisions, which was our primary research objective. Future work could certainly expand to other model architectures, but this would require developing comparable explainability techniques for those architectures.

We have implemented these revisions while maintaining the paper's core contribution: analyzing how individuals express personality traits through their language use, and validating these linguistic patterns against established personality theory through explainability techniques. This approach allows us to understand not just whether we can predict personality from text, but how specific language choices reflect underlying personality traits.

We believe these changes have substantially improved the manuscript while maintaining its core contributions.

Sincerely,

Dr. Daniel Ortiz-Martínez

Department of Mathematics and Computer Science

Universitat de Barcelona

---

## [Editor Report · Decision Letter 1]

Text Speaks Louder: Insights into Personality from Natural Language Processing

PONE-D-24-37721R1

Dear Dr. Ortiz-Martínez,

We’re pleased to inform you that your manuscript has been judged scientifically suitable for publication and will be formally accepted for publication once it meets all outstanding technical requirements.

Kind regards,

Vijaya Prakash Rajanala

Academic Editor

PLOS ONE
---

## [Editor Report · Acceptance letter]

PONE-D-24-37721R1

PLOS ONE

Dear Dr. Ortiz-Martínez,

I'm pleased to inform you that your manuscript has been deemed suitable for publication in PLOS ONE. Congratulations! Your manuscript is now being handed over to our production team.

Kind regards,

on behalf of

Dr. Vijaya Prakash Rajanala

Academic Editor

PLOS ONE